# `ANTIC`: Adaptive Neural Temporal In-situ Compressor

**Sandeep S. Cranganore** [* 1]  **Andrei Bodnar** [* 2]  **Gianluca Galletti** [1 3]  **Fabian Paischer** [1 3]  **Johannes Brandstetter** [1 3]

 **AndreiB137/ANTIC**

## Abstract

The persistent storage requirements for high-resolution, spatiotemporally evolving fields governed by large-scale and high-dimensional partial differential equations (PDEs) have reached the petabyte-to-exabyte scale. Transient simulations modeling Navier-Stokes equations, magnetohydrodynamics, plasma physics, or binary black hole mergers generate data volumes that are prohibitive for modern high-performance computing (HPC) infrastructures. To address this bottleneck, we introduce `ANTIC` (*Adaptive Neural Temporal in situ Compressor*), an end-to-end in situ compression pipeline. `ANTIC` consists of an adaptive temporal selector tailored to high-dimensional physics that identifies and filters informative snapshots at simulation time, combined with a spatial neural compression module based on continual fine-tuning that learns residual updates between adjacent snapshots using neural fields. By operating in a *single streaming pass*, `ANTIC` enables a combined compression of temporal and spatial components and effectively alleviates the need for explicit on-disk storage of entire time-evolved trajectories. Experimental results demonstrate how storage reductions of several orders of magnitude relate to physics accuracy.

## 1. Introduction

As the spatiotemporal resolutions of large-scale high-dimensional scientific computing simulations such as computational fluid dynamics (Ashton et al., 2025, CFD),

---

[1]Institute for Machine Learning, Ellis Unit, JKU Linz, Austria [2]University of Manchester, United Kingdom [3]EMMI AI, Linz, Austria. Correspondence to: Sandeep S. Cranganore <cranganore@ml.jku.at>, Andrei Bodnar <andrei.bodnar@student.manchester.ac.uk>.

*Proceedings of the $43^{rd}$ International Conference on Machine Learning*, Seoul, South Korea. PMLR 306, 2026. Copyright 2026 by the author(s).

plasma physics (Siena et al., 2025; Paischer et al., 2025a), high energy physics (CERN, 2025), weather and climate modeling (Govett et al., 2024; Bodnar et al., 2025b) continue to increase, the generated data volume of transient data has grown from tens of petabytes to exabytes. Historically, storage capacity followed a rapid exponential increase (Kryder & Kim, 2009). However, the pace of storage cost reduction has slowed in recent decades and is no longer advancing at the same rate as compute capability and cost, which continues to evolve under Moore-like scaling. As a result, explosive data growth poses a severe bottleneck for broader adoption and scalability of advanced scientific workflows. For instance, high-fidelity volumetric CFD simulations might contain at least $10,000$ evolution steps with simulation meshes of billions to trillions of volumetric mesh cells (Rossinelli et al., 2013), that result in petabytes of storage requirements (Ashton et al., 2025). Rapidly expanding storage demands are expected to soon exceed the capacity of existing HPC infrastructures (Reed & Dongarra, 2015; Luttgau et al., 2018; Thomas et al., 2021; HPCwire, 2025).

Data reduction in large-scale scientific simulations can be viewed along two complementary axes, (i) the temporal axis, which determines which snapshots to retain, and (ii) the spatial axis, which governs how each snapshot is compressed. Offline (post-hoc) compression is infeasible for large-scale simulations that reach petascale or exascale due to the overwhelming storage demands. Therefore, compression must be performed online (in-situ) as the simulation evolves, avoiding the need to archive the original trajectory. However, most existing in-situ compression methods are not based on the temporal behavior of the underlying physics. Stiff or multi-rate PDE simulations exhibit pronounced multiscale dynamics (Hairer & Wanner, 1996; Gottlieb et al., 2009), where existing temporal subsampling techniques may miss fast transient dynamics or oversample slow and uneventful phases. Furthermore, the nonlinear and multiscale behavior of stiff PDEs exhibit non-stationarity along the spatial axis, which challenges traditional compression schemes based on fixed spatial representations.

To address these critical bottlenecks, we introduce `ANTIC` (*Adaptive Neural Temporal In-situ Compressor*), an *in-situ* framework that addresses both the temporal and spatial

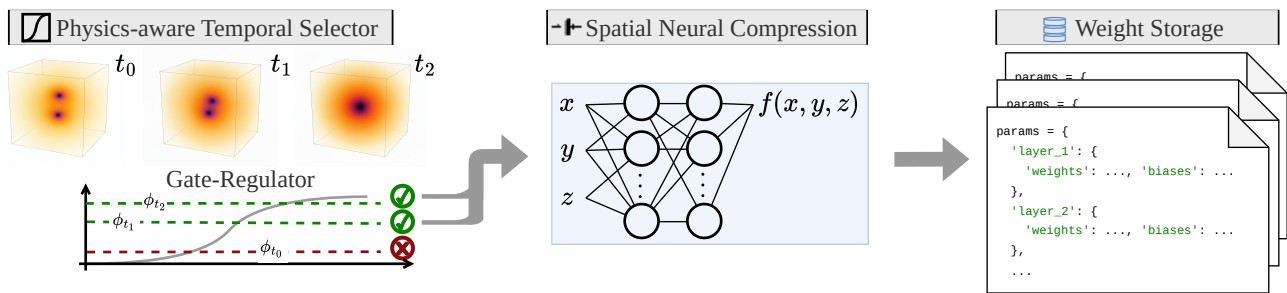

*Figure 1.* **Figure 1: Schematic Overview of ANTIC. ANTIC** facilitates high-fidelity neural compression via a dual-stage asynchronous architecture. **Physics-aware Temporal Selector** serves as a physics-aware filter that identifies salient physical transients. **Spatial Neural Compression** encodes the selected snapshots into compact NN weights, stored in the persistent storage. **ANTIC** enables a combined in-situ compression resulting in multi-fold reduction in data volume while preserving the temporal coherence of the entire simulation.

axes by incorporating (i) a Physics-aware Temporal Selector driven by physics of interest, and (ii) Spatial Neural Compression based on continual fine-tuning (CFT) of neural fields (Takikawa et al., 2023, NFs) to learn residuals between adjacent snapshots. The temporal selector provides an acceptance/rejection decision informed by physics-of-interest of a new snapshot and a context of preceding snapshots. For spatial compression, CFT learns the residuals of a new snapshot to the preceding one in the form of NF weights. Learning these residuals can be done via full fine-tuning or low-rank updates (Hu et al., 2021). By varying the rank of the residual updates, we expose an accuracy-memory Pareto front that can be traversed according to the user's needs.

We evaluate the efficacy of ANTIC across three fundamental axes: (i) *Storage Efficiency*, quantified by the aggregate spatiotemporal compression ratio; (ii) *Spatial Fidelity*, assessing the physics reconstruction quality upon decompression; (iii) *Computational Throughput*, measuring the training-time per temporal snapshot. We consider two physical regimes characterized by multirate dynamics and temporal stiffness (Hairer et al., 2008; Hairer & Wanner, 1996): turbulent 2D Kolmogorov flows and a large-scale 3D Binary Black Hole (BBH) merger simulation comprising $4.2$ TiB of data. On the 2D Kolmogorov turbulence, ANTIC reduces the number of compressed snapshots by $\mathbf{62\%}$ alongside a $\mathbf{47\times}$ spatial compression factor per snapshot, resulting in a net compression of up to $\mathbf{435\times}$. The 3D BBH merger (Pretorius, 2005), serves as a memory-intensive simulation example for which ANTIC achieves $\sim \mathbf{45\%}$ temporal reduction coupled with spatial compression ratios up to $\mathbf{3744\times}$ per snapshot, resulting in spatiotemporal compression of up to $\mathbf{6807\times}$ for the whole trajectory.

To summarize, we make the following contributions:

- We propose ANTIC, an in-situ compression framework for multi-rate/stiff PDE simulations that combines adpative temporal sampling with spatial compression.
- PDE-dependent physics-aware metrics for salient temporal snapshot selection.

- Neural spatial compression based on CFT of residuals between adjacent snapshots.

## 2. Background

In general, there are two different axes along which compression can occur in-situ for stiff PDEs, the spatial and the temporal dimension. Many existing works focus on either of them in isolation or combine them in an offline setup. We elaborate on the different existing techniques below.

### 2.1. Temporal Frame Selection Methods

In the regime of high-fidelity streaming simulations, in-situ keyframe selection is typically mediated by temporal sampling schemes and have emerged as a primary strategy for managing large-scale time-varying volumetric datasets. Existing methodologies focus predominantly on identifying salient snapshots to minimize redundancy for post-hoc visualization and forensic analysis (Larsen et al., 2022; Wu et al., 2022; Tong et al., 2012; Yamaoka et al., 2019; Wang et al., 2008). However, a critical limitation of these heuristic-based selectors is their agnosticism toward the underlying physical invariants of the dynamical system. By decoupling the sampling strategy from the governing physics-based features, these methods can become inefficient when the simulated system changes. Conversely, ANTIC sidesteps this issue by identifying system-specific measures that are sensitive to the physics of interest for different stiff PDEs.

### 2.2. Spatial Compression Techniques

Traditional scientific data reduction techniques primarily rely on transform-based codecs such as JPEG2000 (ISO Central Secretary, 2024), the discrete wavelet transform (DWT) (Kolomenskiy et al., 2022), or leverage low-rank structures through tensor compression (Ballester-Ripoll et al., 2020). While these methods and early autoencoder-based architectures (Le & Tao, 2024) provide significant volume reduction, they often struggle to capture the com-

plex, multiscale correlations inherent in high-dimensional PDE solvers and can become memory costly for lossless compression methods such as FPZIP (Lindstrom & Isenburg, 2006), or ACE (Fout & Ma, 2012).

To address these limitations, recent work has increasingly turned toward Neural Fields (Takikawa et al., 2023; Mildenhall et al., 2021; Müller et al., 2022; Mescheder et al., 2019; Dupont et al., 2021; Jia et al., 2025). This paradigm facilitates Neural Compression (NC) by embedding high-dimensional gridfunctions into the compact, differentiable weights of coordinate-based networks. Such representations offer continuous query access and differentiable modeling, yielding several orders of magnitude compression factors for large-scale, multidimensional scientific simulations, while maintaining high-fidelity physics reconstruction.

NC has been demonstrated across diverse scientific domains: including multidimensional climate modeling (Huang & Hoefler, 2023), relativistic simulations (Cranganore et al., 2025), and plasma physics (Galletti et al., 2026). The latter achieves memory reductions as high as $70,000\times$ using vector quantization (van den Oord et al., 2017) and physics-informed losses.

### 2.3. Parameter-efficient Fine-tuning (PEFTs)

Parameter-Efficient Fine-Tuning (PEFT) (Mangrulkar et al., 2022), specifically Low-Rank Adaptation (LoRA) (Hu et al., 2021), addresses the prohibitive costs of full-parameter updates by exploiting the low intrinsic dimensionality of high-dimensional models (Aghajanyan et al., 2021). This paradigm assumes that weight updates during adaptation reside in a low-rank manifold. Consequently, for a frozen pre-trained weight matrix $\mathbf{W}_0 \in \mathbb{R}^{n \times m}$, the updated weights $\mathbf{W}$ are expressed as an outer product $\mathbf{B}\mathbf{A}$ where $\mathbf{A} \in \mathbb{R}^{r \times m}$ and $\mathbf{B} \in \mathbb{R}^{n \times r}$ are trainable low-rank factors with rank $r \ll \min(m, n)$. LoRA introduces zero latency during inference, as the product $\mathbf{B}\mathbf{A}$ can be merged directly into the pre-trained weights. This constrained optimization framework has demonstrated the ability to match or exceed the accuracy of full fine-tuning as shown in Xin et al. (2024).

Recently, Truong et al. (2025) demonstrated a versatile instance-specific neural field editing via LoRA, which merely requires lightweight updates to pre-trained NF parametrizations for visual data processing and geometry processing tasks (see also Kang et al. (2024); Liu et al. (2021) for other usecases of PEFT in MLPs). Within the compression community, Galletti et al. (2026) stabilized physics-informed training of large autoencoders using EVA (Paischer et al., 2025b). We consider LoRA-style fine-tuning to establish a accuracy-complexity trade-off to provide a flexible in-situ framework based on user demands.

### 2.4. Codecs & (Neural) Video Compression

Codecs are generally considered as post-hoc or offline compression that combine both temporal and spatial axes. In an offline setup, many aforementioned traditional compression methods can be applied by collapsing the temporal axis into the spatial ones. However, this is infeasible for large-scale scientific simulations due to storage demands as a single trajectory may comprise hundreds of terabytes of data or beyond. Therefore, our setup necessitates online or in-situ compression. Nonetheless, we briefly elaborate on relevant work on offline compression and codecs.

Video compression standards, such as H.264 (AVC) (Richardson, 2010), H.265 (HEVC) (Ohm et al., 2012), and VVC (H.266) (Bross et al., 2021), achieve state-of-the-art efficiency by exploiting spatiotemporal redundancies through motion-compensated prediction and residual transform coding. By partitioning frames into adaptive Coding Tree Units (CTUs) and leveraging inter-frame dependencies, these codecs reduce the bitrate of high-dimensional visual data by multiple orders of magnitude. The emergence of Neural Video Compression (NVC) frameworks (Lu et al., 2019) has further advanced this field, achieving rate-distortion performance competitive with, or exceeding, traditional codecs.

Recent advances have gravitated toward the integration of NVC with adaptive keyframe (snapshot) selection to optimize scene motion dynamics. For instance, Jha et al. (2025) employ momentum-aware thresholding to dynamically curate informative frames, while Zhang & Gao (2025) utilize learned rate-control mechanisms to prioritize frames characterized by high information entropy or significant viewpoint shifts. While these computer-vision-centric methods excel at reconstructing 3D scene dynamics for human perception, they fail for scientific computing applications, as they do not consider physics of interest.

Prior spatiotemporal adaptive frameworks like MGARD (Gong et al., 2023a;b) modulate numerical precision through feature-aware error bounds while maintaining a uniform temporal output frequency and feature-driven compression. In contrast, ANTIC performs non-uniform temporal selection, leveraging physical saliency (in-situ physics metrics/invariants extraction) to bypass redundant snapshots entirely in-situ, retaining only high activity snapshots. Furthermore, our NF approach is flexible enabling continual fine-tuning using low-rank schemes and achieves compression ratios significantly exceeding those of classical lossy, error-bound compressors ($\text{Compression}_{\text{NF}} \gg \text{Compression}_{\text{classical}}$) at equivalent fidelity for time-evolved simulations.

Adaptive Neural Temporal In-situ Compressor

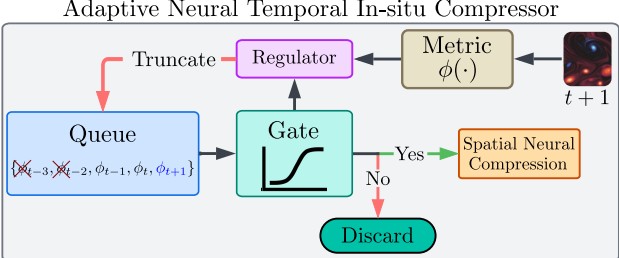

*Figure 2.* Workflow of ANTIC. A new snapshot at time $t + 1$ is passed over by the simulator. The Metric extracts physics of interest for the new snapshot $\phi_{t+1}$, which is passed to the Regulator. Based on $\phi_{t+1}$, the regulator truncates the context queue and adds $\phi_{t+1}$. Finally, the truncated context is passed to the gate along with $\phi_{t+1}$ to determine whether or not to compress the new snapshot.

## 2.5. In-situ Compression

Other *in-situ* data compression frameworks of spatiotemporal simulation data include Glaws et al. (2020); Balin et al. (2023) for large-scale computational fluid dynamics (CFD), and gyrokinetic tokamak simulations (Lakshminarasimhan et al., 2011, ISABELA). Although each of these methodologies lack either an adaptive temporal selection criteria or NC. As a result, these methods offer modest compression rates ($\sim 7\%$), or are based on autoencoders with fixed-sized latents that are not resolution invariant. ANTIC differs as it combines a temporal selector that is tailored toward physics-of-interest and by providing a accuracy-memory pareto front for spatial compression that can be tailored to the user's needs.

## 3. Methodology

ANTIC consists of two core components, (i) Physics-aware Temporal Selector (PATS), and (ii) spatial neural compression that is postulated as traversing a rate-distortion pareto front. We eleborate on the PATS and its core parts in Section 3.1. The spatial neural compression is explained in detail in Section 3.2. The corresponding pseudocode for ANTIC is detailed in Algo. 3.

### 3.1. Physics-aware Temporal Selector

The PATS is realized as a non-parametric modular architecture, consisting of four vital parts (see Figure 2), namely (i) a Metric, (ii) a Regulator, (iii) a Queue, (iv) and the Gate.

**Queue.** The Queue stores the context that is necessary for the Gate to provide a decision on whether to compress the current snapshot or not. It is represented as a sliding window of size $W$. Importantly, the Queue only maintains the extracted physics metric $\phi(\cdot)$ for each snapshot in the window $\{t_1, t_2, \ldots, t_W\}$. This facilitates the detection of phase transitions captured by the Metric and hence provides

vital information on whether or not to compress a snapshot.

**Regulator.** After a phase transition is detected, it is vital to rearrange the content of the Queue as the stored snapshot metrics may interfere with each other. The Regulator dynamically modulates the window size $W$ (variable stopping within $W$) of the Queue, ensuring the selection procedure remains invariant to non-stationary shifts in the characteristic time-scales of a simulation. Thus, our framework is based on a *dynamic regulator*. Once a snapshot $t^*$ is selected and compressed, $W$ is truncated such that $t^*$ becomes the new reference anchor, effectively resetting the temporal origin (clears memory of previous steps) for the subsequent selection window.

**Metric.** The Metric is the core component that induces a notion of physics-awareness. Depending on the underlying PDE system, the Metric is instantiated with different measures. For fluid dynamics, for example, phase transitions may be characterized via the enstrophy signal (Doering & Gibbon, 1995), whereas for gravitational wave modeling it might be instantiated as the magnitude of the Weyl scalar (Teukolsky, 1973; Hinder et al., 2011; Iozzo et al., 2021). Based on the underlying PDE, this function is computed for each new snapshot that arrives from the simulation, where it is propagated to the Regulator.

**Gate.** This unit functions as a local decision engine designed to resolve transient physical phenomena that escape fixed-interval selection schemes. It receives the current truncated context from the Queue and the physics quantity from the Metric to form a dynamic threshold. This threshold has two use-cases. First, it facilitates the final decision on whether or not a new snapshot should be compressed. Secondly, it serves as a feedback signal to the Regulator, enabling it to adaptively modulate the window size for the next snapshot.

### 3.2. Spatial Neural Compression

To enable effective spatial compression, we adopt a CFT–based adaptation strategy using neural fields. For many PDEs, successive solution states evolved at different time intervals $\Delta t$ differ primarily by smooth, low-magnitude perturbations governed by the underlying dynamics. Rather than compressing each state independently, these perturbations can be naturally captured by fine-tuning a neural field initialized at time $t$ to learn an implicit representation at $t + \Delta t$. This procedure simplifies the learning task as only the residual between adjacent snapshots needs to be encoded in the neural field weights.

Through this lens, fine-tuning can be re-interpreted as continually learning a residual update to a pretrained neural field as the simulation evolves. Specifically, we define perturbative temporal changes between successive snapshots

as $u(t + \Delta t) - u(t) \approx \Delta u(t)$. The perturbative update $\Delta u(t)$ is encoded in an incremental update to the weights of a neural field as $\mathbf{W}_{t+\Delta t} = \mathbf{W}_t + \Delta \mathbf{W}_{\Delta t}$.

Leveraging LoRA as a fine-tuning strategy, we can reparameterize the perturbative weight update as $\Delta \mathbf{W}_{\Delta t} = \mathbf{A}^{(\Delta t)} \mathbf{B}^{(\Delta t)}$, where $\mathbf{A} \in \mathbb{R}^{n \times r}$ and $\mathbf{B} \in \mathbb{R}^{r \times k}$ are trainable matrices of rank $r \ll \min(n, k)$. This reparameterization induces a natural trade-off between compression rate and reconstruction accuracy. Restricting the update capacity, i.e., the rank of $\mathbf{A}^{(\Delta t)} \mathbf{B}^{(\Delta t)}$, limits the number of parameters required to encode the residual, and hence reduces the memory footprint, but may increase distortion, whereas full fine-tuning reduces distortion at the cost of a larger memory footprint. By transitioning between LoRA and full fine-tuning, we obtain a family of models that span an accuracy–compression (rate–distortion) Pareto frontier.

## 4. Experiment Setup

Our two main experiments constitute the 2D Kolmogorov flow and 3D Binary Black Hole (BBH) merger. The 2D Kolmogorov flow serves as a stress-test for our PATS; its multi-rate eddy dynamics and localized Reynolds number fluctuations provide a rigorous validation for non-uniform sampling (Hairer et al., 2008). The 3D BBH simulation represents our primary stress test for extreme spatial NC. Given the high-resolution volumetric tensor fields and the 4D nature of the simulation, it provides an ideal testbed for investigating the accuracy-memory trade-off across our proposed training paradigms.

**2D Kolmogorov flow simulations.** This benchmark serves as a controlled environment to validate the PATS module's capacity to resolve high-wavenumber components and transient spatiotemporal correlations. Starting from high-frequency initial conditions (Dresdner et al., 2022), the system evolves through a transition to decaying turbulence where small-scale vortices coalesce. The data generation pipeline is based on $\mathtt{JAX - CFD}$ (Dresdner et al., 2022). The memory-per-snapshot is **17 MiB**, yielding **16.8 GiB** for an entire trajectory of 1000 time-steps. The data preparation specifics is detailed in Appendix B.1. We select **enstrophy** as the physics metric for the decaying turbulence fluid, which is defined as

$$\mathcal{E}(t) = \frac{1}{2} \int_{\Omega \subset \mathbb{R}^2} ||\omega(x, y, t)||^2 dA, \tag{1}$$

where $\omega$ is the scalar vorticity field over a 2D domain $\Omega$. Enstrophy encodes the magnitude of turbulence, therefore it is an ideal proxy to determine fast/slow evolving regions to inform our PATS.

**3D Binary Black Hole (BBH) Mergers.** The 3D BBH merger describes a high-dimensional dynamical system in which two black holes evolve and merge within a learned geometry, while emitting gravitational waves that encode and transmit information about the system's changing internal state. The underlying system is governed by the Einstein Field Equations (EFEs), a system of highly nonlinear, second-order hyperbolic-elliptic tensor-valued PDEs that describes the gravitational field as distortions of the 4D spacetime geometry induced by matter. As analytic solutions are restricted to idealized symmetries, Numerical Relativity (NR) is required to model complex astrophysical phenomena (Abbott et al., 2016a;b). We employ the Baumgarte-Shapiro-Shibata-Nakamura (BSSN) formalism, a stable $3 + 1$ decomposition (Baumgarte & Shapiro, 2010; Shibata & Nakamura, 1995; Hayashi et al., 2025) that evolves volumetric scalar, vector, and tensor fields (the BSSN variables) over time. The simulation stream is generated via $\mathtt{JAX\_NR}$ (Bodnar et al., 2025a), a GPU-accelerated NR solver. Each temporal snapshot comprises 18 evolved variables totaling **0.7 GiB**, with a complete trajectory (5, 966 steps) yielding **4.2 TiB** of raw data. Refer to Section B.2 for a formal derivation of the BSSN system and data preprocessing protocols. In gravitational wave detection, the Weyl scalar $\Psi_4(t, \mathbf{r})$ (Teukolsky, 1973; Iozzo et al., 2021; Hinder et al., 2011) extracted at a radius $\mathbf{r}$ is a crucial complex-valued scalar quantity that describes the outgoing transverse radiation field used in extracting waveforms from NR simulations. Simply put, the Weyl scalar measures how the outgoing gravitational wave field develops and propagates over time, isolating the merging event of the two black holes. Therefore, we instantiate our metric with it to facilitate online salient snapshot selection.

### 4.1. NF Implementation Details

The NC module employs a coordinate-based MLP with SiLU activations (Elfwing et al., 2018) as a base model. The architecture of the base model is based on **256 × 6** (hidden size × number of layers), amounting to $\sim 0.4$M parameters. To mitigate spectral bias and resolve high-vorticity transients in the Kolmogorov flows, we employ Fourier Feature Mapping (FFM) (Tancik et al., 2020) with an embedding dimension of 256. We use the SOAP optimizer (Vyas et al., 2025), a second-order preconditioner, with a cosine annealing scheduler (Loshchilov & Hutter, 2017). We initially observed numerical instabilities of CFT arising from exploding weight magnitudes. To mitigate this we employ LayerNorm (Ba et al., 2016), which stabilizes the distribution of intermediate activations across continual weight updates, and weight decay (Loshchilov & Hutter, 2019), which penalizes large weight magnitudes and prevents unbounded growth of the parameter norm across successive snapshots. For training and CFT of the base model (Scratch) the learning rate is annealed from $10^{-3}$ to $10^{-5}$. If CFT is instantiated via LoRA, we use a higher initial rate of $10^{-2}$ following the recommendations of Hayou et al. (2024). Fur-

*Table 1.* **Performance of ANTIC compared to baselines on 2D Kolmogorov flows and 3D BSSN simulation trajectories**. We compare (i) Uniform sparse sampling (every 5th step), (ii) Dense sampling (every timestep), combined with (i) traditional compression (ZFP), (ii) CFT, or (iii) CFT via LoRA fine-tuning. For comparison to ZFP we match reconstruction quality to ANTIC and compare for compression ratio. TR (Temporal Retention) denotes the percentage of the simulation duration preserved and Physics-awareness (PA) denotes whether or not the temporal selector uses a PDE specific metric. SC (Spatial Compression) and TC (Total Compression) represent the memory reduction factors. ANTIC consistently outperforms ZFP and different temporal sampling schemes. In Sec. F, we report the performance of neural compression (CFT, CFT+LoRA) against well-known scientific computing classical compressors like SZ3, MGARD, Blosc-2 zstd, clearly indicating the extreme compression obtained via neural fields for 3D high-resolution tensor-valued BSSN evolution simulations.

| METHOD | 2D KOLMOGOROV FLOWS (16.8 GIB) | | | | 3D BSSN SIMULATIONS (4.2 TIB) | | | |
|---|---|---|---|---|---|---|---|---|
| | TR($\uparrow$) (%) | PA ($\checkmark$/$\times$) | SC($\uparrow$) ($\times$) | TC($\uparrow$) ($\times$) | TR($\uparrow$) ($\times$) | PA ($\checkmark$/$\times$) | SC($\uparrow$) ($\times$) | TC($\uparrow$) ($\times$) |
| SPARSE SAMPLING + ZFP [TOL=1e-1] | 20% | $\times$ | 13$\times$ | 65$\times$ | 20% | $\times$ | 27$\times$ | 141$\times$ |
| DENSE SAMPLING + ZFP [TOL=1e-1] | 100% | $\checkmark$ | 13$\times$ | 13$\times$ | 100% | $\checkmark$ | 27$\times$ | 28$\times$ |
| PATS + ZFP [TOL=1e-1] | 37% | $\checkmark$ | 13$\times$ | 120$\times$ | 55% | $\checkmark$ | 27$\times$ | 52$\times$ |
| SPARSE SAMPLING + FT | 20% | $\times$ | 12$\times$ | 60$\times$ | 20% | $\times$ | 471$\times$ | 2457$\times$ |
| SPARSE SAMPLING + LoRA | 20% | $\times$ | 47$\times$ [r: 32] | 235$\times$ | 20% | $\times$ | 3744$\times$ [r: 16] | 18720$\times$ |
| DENSE SAMPLING + FT | 100% | $\checkmark$ | 12$\times$ | 12$\times$ | 100% | $\checkmark$ | 471$\times$ | 471$\times$ |
| DENSE SAMPLING + LoRA | 100% | $\checkmark$ | 47$\times$ [r: 32] | 47$\times$ | 100% | $\checkmark$ | 3744$\times$ [r: 16] | 3744$\times$ |
| ANTIC-FT (OURS) | 37% | $\checkmark$ | 12$\times$ | 111$\times$ | 55% | $\checkmark$ | 471$\times$ | 860$\times$ |
| ANTIC-LoRA (OURS) | 37% | $\checkmark$ | 47$\times$ [r: 32] | 435$\times$ | 55% | $\checkmark$ | 3744$\times$ [r: 16] | 6807$\times$ |

thermore, we always sweep over LoRA ranks. By default, all our training runs are conducted in FLOAT32 precision. Detailed hyperparameter configurations for both 2D Kolmogorov and 3D BSSN simulations are provided in Appendix B.1.1 and B.2.1.

## 5. Results

The performance of ANTIC is gauged by validating the combined performance of its two core modules across disparate physical regimes: (i) the PATS detailed in Sec. 3.1 for resolving transient dynamics, and (ii) the NC for spatial compression as described in Sec. 3.2. We present the results in detail for each usecase. Moreover, we show for the 2D Kolmogorov flows that our methods' high-fidelity global reconstruction capabilities persist even over long-horizons, detailed in Sec. G.1.

### 5.1. 2D Kolmogorov flow simulations

**Main results.** We report results for two variants of ANTIC, namely (i) ANTIC-FT, which always performs full fine-tuning for CFT, and (ii) ANTIC-LoRA, which uses low-rank adaptation instead. We compare our ANTIC variants to various temporal sampling approaches (Sparse or Dense) and to a classical compression method, namely ZFP (Lindstrom, 2014). All methods are tuned to reach a similar and acceptable level of reconstruction quality (relative $\ell_2 \sim 10^{-3}$) and we compare the resulting compression ratios. To evaluate the different approaches, we report total compression (TC),

which combines temporal retention (TR) with spatial compression (SC) in Table 1. Our proposed variants achieve the highest temporal coherence, consistently outperforming both Sparse (every 5th snapshot) and Dense (every snapshot) baselines. Specifically, compared to ZFP, ANTIC provides 4$\times$ greater spatial compression. Furthermore, due to the PATS module we successfully retain meaningful transients as shown in Fig. 3 (left) which amounts to 37% TR of the entire trajectory. In contrast, Dense sampling attains 100% TR, but results in significantly lower TC as it compresses every snapshot. Vice-versa, Sparse sampling results in low (20%) TR, but high TC, however, neglects plenty of informative transients. In terms of SC, FT and LoRA attain comparable or higher compression rates as ZFP. We also refer the reader to Sec. F for comparison against other scientific computing classical compression algorithms.

High-fidelity global reconstruction of turbulence-laden fluid simulations, including the faithful capture of local physical structures, is also a core requirement of ANTIC. To substantiate that ANTIC consistently achieves a relative $\ell_2$ error in the $10^{-3}$–$10^{-4}$ range, we evaluate reconstruction quality against the ground truth through three complementary diagnostics: the enstrophy flux signal $\mathcal{E}(t)$, the maximum absolute pointwise error, and the vorticity field $\omega(\mathbf{x}, t) \, \forall \mathbf{x} \in \mathbb{R}^2$ at representative timesteps spanning the peak turbulent regime. Reconstruction plots and quantitative results are reported in Sec. G.1.

**PATS module contribution.** Fig. 3a illustrates the efficacy of PATS on the 2D Kolmogorov flow, which aggressively

sparsifies snapshots during the decaying turbulence phase ($T > 10$). Furthermore, it isolates non-stationary transients, reducing temporal retention to $37\%$, without compromising physical fidelity. The necessity of the PATS submodule for physics-informed, in-situ salient snapshot selection is substantiated in Section H, where we demonstrate that physics-agnostic temporal selectors (for e.g. visual-computing/3D scene reconstruction) such as **momentum-aware** key-frame selector (Jha et al., 2025) or even information-theoretic (cf. (Yamaoka et al., 2019)) **entropy based** selectors: *Jensen-Shannon divergence, residual differential entropy, spectral and normalized mutual information* systematically fail to identify dynamically critical transients in multi-scale, high-dimensional PDE simulation trajectories, thus making them unsuitable for realistic large-scale simulations.

**NC module contribution.** We present results for different training strategies, namely (i) training a NF from scratch for each snapshot, (ii), continual fine-tuning (CFT), and (iii), continual fine-tuning via LoRA ($r \in \{8, 16, 32, 64\}$). We sweep model sizes from $32 \times 2$ (40 KiB) up to $512 \times 8$ (7.5 MiB). The Pareto-frontier in Fig. 4a (left) plots the mean rel $\ell_2$ over five subsequent snapshots in the high-enstrophy regime for the best performing model size $512 \times 8$ (an extended version is detailed in Section D). Interestingly, CFT is $30\%$ more accurate than training from scratch. Moreover, LoRA ($r = 32$) achieves accuracy on-par with CFT while reducing parameter storage by $4\times$. Furthermore, LoRA ($r = 64$) defines the state-of-the-art Pareto front, yielding a $30\%$ improvement in Mean Rel. $\ell_2$ and MAE over CFT. This configuration achieves a $23\times$ compression factor relative to the $17$ MiB ground-truth per snapshot. We provide Rel. $\ell_2$ and MAE for all configurations in Table 2.

*Table 2.* **Accuracy metrics vs memory footprints for 2D Kolmogorov flows.** Different spatial compression schemes evaluated across five sequential snapshots in a highly-turbulent regime. The last column reports the memory compression per snapshot relative to the 17 MiB ground truth. Continual fine-tuning using LoRA establishes an accuracy-memory pareto front.

| Method | Mean Rel. $\ell_2$ (5 snapshots) | Mean MAE (5 snapshots) | Mem. /snapshot |
|---|---|---|---|
| SCRATCH | 1.7e-3 ± 8.5e-5 | 0.20 ± 0.026 | 1.50 MiB (12×) |
| FULL FT | 6.1e-4 ± 8.0e-6 | 0.061 ± 0.011 | 1.50 MiB (12×) |
| LoRA 8 | 6.9e-2 ± 6.3e-3 | 0.63 ± 0.09 | 98 KiB (188×) |
| LoRA 16 | 2.1e-2 ± 2.5e-3 | 0.13 ± 0.02 | 196 KiB (94×) |
| LoRA 32 | 6.8e-4 ± 7.0e-5 | 0.063 ± 0.011 | 392 KiB (47×) |
| LoRA 64 | **4.3**e-4 ± **2.4e-5** | 0.041 ± 0.006 | 784 KiB (23×) |

## 5.2. 3D BBH merger simulations

**Main results.** We report the results for the 3D BBH merger trajectory in Table 1. For this use-case, our proposed variants, ANTIC-FT, LoRA achieve extreme compression ratios

*Table 3.* **Accuracy metrics vs memory footprints for 3D BBH merger simulations.** Different spatial compression schemes evaluated across five sequential snapshots within the black hole merger regime. For completenees, we report wallclock-time per snapshot in the last column for the different training schemes.

| Method | Mean Rel. $\ell_2$ (5 snapshots) | Mem. / snapshot | Time / snapshot |
|---|---|---|---|
| SCRATCH | 6.2e-4 ± 4.2e-5 | 1.52 MiB (471×) | ∼ 162 (s) |
| FULL FT | **4.6e-4 ± 7.6e-5** | 1.52 MiB (471×) | ∼ 162 (s) |
| LoRA 8 | 6.0e-4 ± 4.1e-5 | **98 KiB (7489×)** | ∼ 131 (s) |
| LoRA 16 | 5.7e-4 ± 3.9e-5 | 196 KiB (3744×) | ∼ 133 (s) |
| LoRA 32 | 5.3e-4 ± 3.7e-5 | 392 KiB (1872×) | ∼ 137 (s) |
| LoRA 64 | 4.6e-4 ± 3.2e-5 | 784 KiB (936×) | ∼ 149 (s) |

($3410\times$), and high temporal coherence on the complex 3D relativistic simulations. Importantly, this compression ratio is achieved while maintaining reconstruction quality of relative $\ell_2 \in [10^{-5}, 10^{-4}]$. Again, ANTIC consistently outperforms both Sparse and Dense baselines. Due to the Weyl-scalar metric used in PATS, we successfully retain the activity near the merger phase ($T \in [150M - 200M]$) shown in Fig. 3b, yielding a $55\%$ TR, due to pre/post merger sparsification. In terms of spatial compression, our method offers a significant data volume reduction between ($17\times$ - $65\times$) compared to PATS+ZFP.

The Weyl scalar serves as the primary observable for characterizing the amplitude, frequency, and phase evolution of gravitational waves emitted during compact binary mergers. High-fidelity "global reconstruction" of $\Psi_4$ (we show the magnitude alone) at a fixed extraction radius is therefore essential for post-hoc gravitational wave analysis. As demonstrated in Sec. G.2, ANTIC achieves a relative $\ell_2$ error of $10^{-4}$–$10^{-5}$ and a maximum absolute error in $|\Psi_4(t/M)|$ of $\sim 10^{-5}$ from *in-situ* neural-field-parametrized snapshots. The BSSN-evolved lapse function $\alpha$ (Sec. B.2) achieves a consistent relative $\ell_2$ error of $10^{-4}$–$10^{-5}$ across the spatial domain, with the exception of the black hole interior region; this region requires numerical excision and is appropriately excluded from the error evaluation.

**PATS module contribution.** Fig. 3b shows that PATS using the Weyl-scalar guided metric attains temporal filtering of **55%** retention with dense sampling between $T \in [145M, 185M]$ (merger phase), and reduced sampling during pre-merger (until $T = 140M$) and postmerger phases ($T > 210M$).

**NC module contribution.** We again sweep the same model sizes as for Kolmogorov flows and show the resulting pareto fronts in Fig. 4b for the best model of size $512 \times 8$ (an extended version is detailed in Section D). Notably, CFT and LoRAs ($r : 8, \cdots, 64$) clearly outperform training from scratch. For an accuracy range of $[10^{-4}, 10^{-5}]$ one can

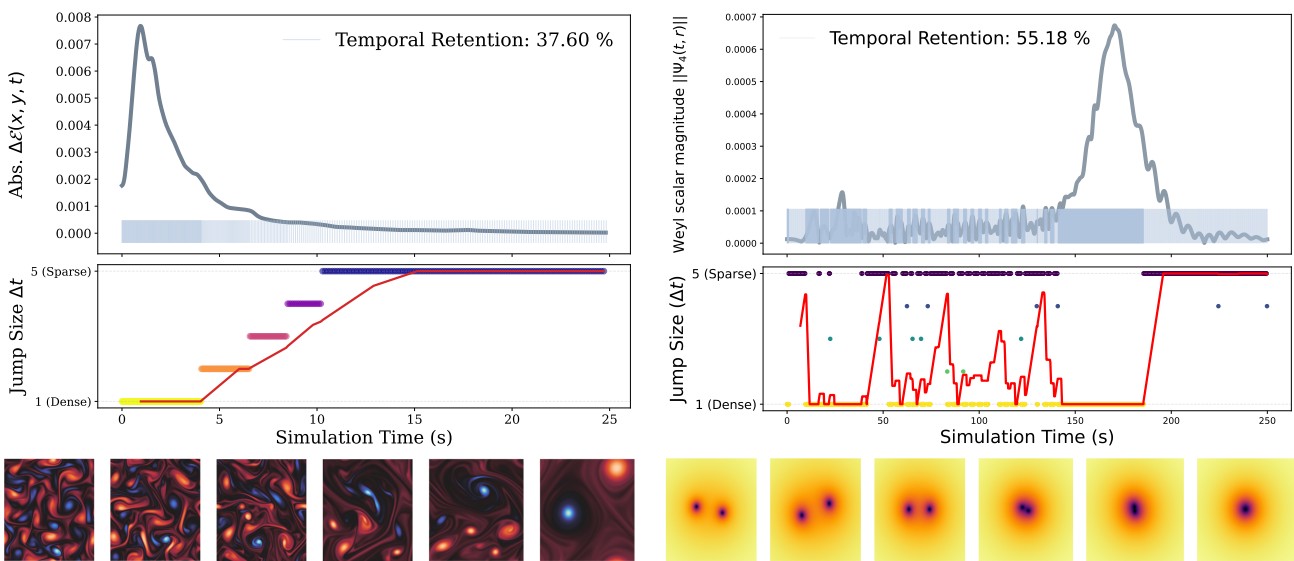

*Figure 3.* **Qualitative results for Physics-aware Adaptive Temporal Selection.** (Left) The enstrophy flux (top) identifies turbulent regions in 2D Kolomogorov flows, hence is well suited as metric for our PATS. As a consequence, PATS densely samples at the start of the trajectory that exhibits strong chaotic behavior, whereas sampling more sparsely in non-turbulent regions. (Right) The Weyl scalar (top) isolates the non-linear gravitational wave (GW), resulting in dense sampling around the merger region and sparse sampling otherwise.

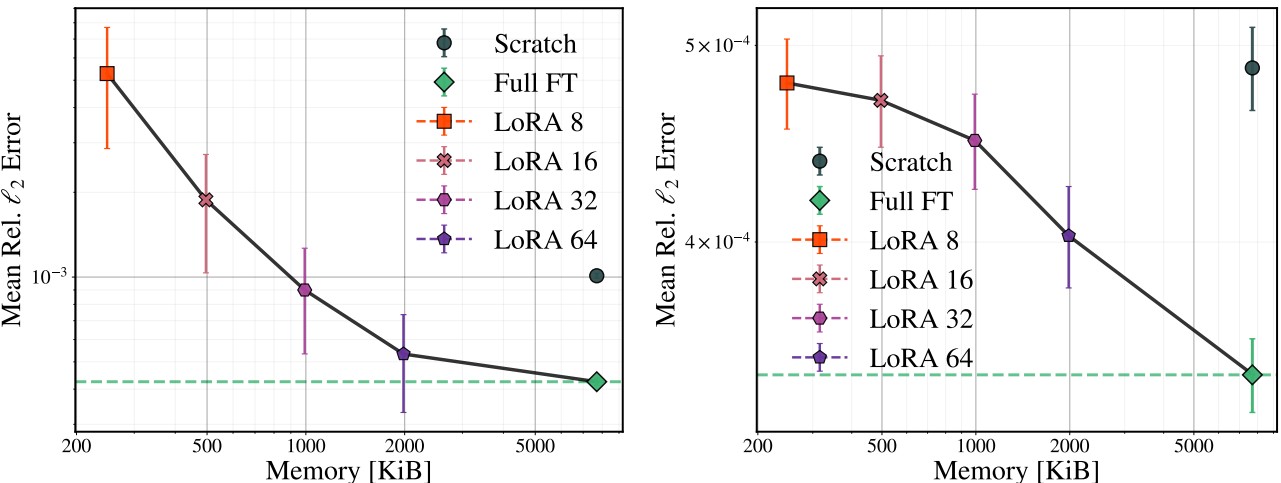

*(a)* **2D Kolmogorov flow pareto-front for spatial compression.** Averaged relative $\ell_2$ error over 5 subsequent frames in highly-turbulent window as a function of parameter memory (KiB). We compare three training paradigms: (i) compressing each snapshot from scratch, (ii) continual fine-Tuning via full fine-tuning (FT), and (iii) via LoRA ($r \in [8, 64]$). As the trendlines demonstrate, LoRA ($r > 32$) attains a comparable error to FT while significantly improving compression rate.

*(b)* **3D BBH merger pareto-front for spatial compression.** Averaged relative $\ell_2$ error over 5 subsequent frames in the near merger regime as a function of parameter memory (KiB). We compare three training paradigms: (i) compressing each snapshot from scratch, (ii) continual fine-Tuning via full fine-tuning (FT), and (iii) via LoRA ($r \in [8, 64]$). LoRA-edited NFs (ranks $r \geq 8$) attain a comparable error to FT while significantly improving compression rate.

achieve an extreme compression factor relative to the raw simulation ground truth of size **0.7 GiB** per BSSN-evolved snapshot, amounting up to **3744×**. Detailed metrics for the base model, including rel. $\ell_2$ and training-time, are provided in Table 3. Moreover, the extreme data-volume reduction capabilities of neural fields compared to traditional scientific computing compressors becomes pronounced for such multi-

dimensional high-resolution mesh-intensive snapshots as detailed in Sec. F.

## 5.3. Neural Compression Scalability and In-Situ Feasibility

Traditional PDE solvers scale super-linearly with spatial resolution, as advancing a single timestep $\Delta t$ requires applying numerical stencils and satisfying CFL stability constraints across all $N^d$ mesh points, with wall-clock costs growing as $\mathcal{O}(N^d)$ or worse under adaptive mesh refinement. Neural field training, by contrast, operates on a fixed-capacity implicit network whose parameter count is decoupled from grid resolution, yielding sub-linear growth in training overhead with increasing resolution. We analyze this scaling behavior in detail for CFL-constrained 2D solvers and benchmark this in Sec. I. Furthermore, neural fields admit patch-based training over disjoint spatial subdomains, offering an additional avenue for acceleration in mesh-intensive use cases.

Importantly, Sec. E and Table 4 demonstrate that CFT-based neural compression significantly reduces per-snapshot training time by learning only residual updates between successive snapshots. For our main use case, i.e. the 3D binary black hole merger simulations — CFT converges to a per-snapshot MSE of $\sim 10^{-8}$ in approximately $1/10^{\text{th}}$ the epochs required by cold-start parametrization, reducing per-snapshot training time from $\sim 45$ s to $\sim 4.5$ s (a speedup exceeding $10\times$). This demonstrates that CFT exploits inter-snapshot continuity to reduce the per-snapshot encoding cost from $\mathcal{O}(T_{\text{cold}})$ to $\mathcal{O}(T_{\text{CFT}}) \ll T_{\text{cold}}$.

Together, the resolution-decoupled scaling of neural field training and the inter-snapshot efficiency of CFT establish that neural compression is a viable and practical paradigm for *in-situ* integration with large-scale, mesh-intensive PDE simulations, reducing the throughput gap between the solver and the training.

## 6. Limitations.

**(i) Computational Latency.** Despite the CFT scheme reducing per-snapshot neural compression wall-clock time by $> 10\times$ relative to cold-start parametrization, the solver usually remains an order of magnitude faster than neural field training (Sec. E). While the sub-linear scaling of neural compression relative to CFL-constrained solvers progressively closes this gap at high resolutions (Sec. I), achieving real-time *in-situ* deployment will require further reductions in per-snapshot training latency. **(ii) Derivative Fidelity.** Standard MLPs lack the inductive bias necessary to faithfully preserve high-order spatial derivatives, which are critical for downstream physics analysis and automatic differentiation pipelines. This limitation can be partially addressed through Sobolev-space supervision via higher-order derivative losses (Czarnecki et al., 2017), though at the cost of additional training overhead. **(iii) Domain-Specific Temporal Metrics.** The physics-based signal powering PATS is necessarily tailored to the underlying PDE system; for instance, enstrophy flux for turbulent flows and the Weyl scalar for spacetime evolution. We demonstrated that this specificity confers clear advantages over physics-agnostic selectors (Sec. H), however the construction of generalizable, PDE-agnostic temporal selectors that retain sensitivity to dynamically critical transients remains an open problem, which we defer to future investigation. **(iv) Off-Grid Generalization.** The present study trains and evaluates ANTIC on the simulation grid, optimizing the relative $\ell_2$ error over grid-coincident coordinates. Generalization to arbitrary off-grid query points, enabled by the continuous implicit representation of the neural field, remains to be systematically validated and constitutes a natural extension of this work.

## 7. Conclusion

We have introduced ANTIC, an end-to-end *in-situ* neural compression pipeline paving the way for extreme compression for large-scale scientific simulations. ANTIC combines a physics-aware temporal selector, tailored to the dynamical structure of high-dimensional PDE trajectories, with a spatial neural compression module that is based on continual fine-tuning (CFT) of neural fields on salient snapshot. CFT reduces per-snapshot encoding cost by exploiting inter-snapshot continuity, requiring only perturbative weight updates between subsequent neural fields, yielding a $10\times$ reduction in training wall-clock time relative to cold-start methodologies. ANTIC integrates seamlessly into complex simulation streams, achieving compression ratios exceeding $400\times$ for turbulent 2D Kolmogorov flows and $10,000\times$ for 3D BSSN evolved binary black hole merger simulations, while maintaining spatial fidelity across both use cases. It equips practitioners with a principled and computationally efficient tool for the *in-situ* storage of high-dimensional scientific simulation data as problem scales and physical complexity continue to grow.

**Future Work.** We believe that our work opens up fruitful and insightful future research topics. (i) *Integration.* As a first step, we aim for deployment of ANTIC into HPC-scale simulation pipelines in computational fluid dynamics (Nielsen et al.; Rossinelli et al., 2013; Jia et al.) or other compute and memory intensive simulations such as gyrokinetics or astrophysics. This includes data and pipeline parallelism for decreasing wall-clock training time related bottlenecks of neural spatial compression. (ii) *Learning temporal selection.* Our Physics-aware Temporal Selector ultimately constitutes a binary decision making process as the simulation evolves. In the future we aim to investigate whether we can learn a decision making policy via reinforcement learning to enhance universality.

## Impact Statement

This work addresses a growing bottleneck in large-scale scientific computing by reducing the storage and data-management burden of high-resolution PDE simulations. By enabling efficient in-situ compression of spatiotemporally evolving fields, the proposed method can make large-scale simulations more accessible to researchers with limited storage resources and reduce the energy and infrastructure costs associated with them. Potential downstream applications span climate modeling, fluid dynamics, plasma physics, and astrophysics, where long-running simulations are essential for scientific discovery. The method is designed for scientific data compression and does not introduce new risks beyond those already present in numerical simulation and data-driven modeling; however, care must be taken to ensure that compression-induced errors remain acceptable for downstream analysis.

## Acknowledgements

The ELLIS Unit Linz, the LIT AI Lab, the Institute for Machine Learning, are supported by the Federal State Upper Austria. We thank the projects FWF AIRI FG 9-N (10.55776/FG9), AI4GreenHeatingGrids (FFG-899943), Stars4Waters (HORIZON-CL6-2021-CLIMATE-01-01). We thank NXAI GmbH, Audi AG, Silicon Austria Labs (SAL), Merck Healthcare KGaA, GLS (Univ. Waterloo), TÜV Holding GmbH, Software Competence Center Hagenberg GmbH, dSPACE GmbH, TRUMPF SE + Co. KG.

Sandeep S. Cranganore was supported by the FWF Bilateral Artificial Intelligence initiative under Grant Agreement number 10.55776/COE12.

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

# A. Physics-aware Temporal Selector suite

Temporal selection criteria vary across domains, depending on the underlying PDE and the transient dynamics dictated by the system's intrinsic physical phenomena. In our framework, specific physical quantities extracted from the simulation act as a decision engine for identifying salient snapshots. To build a robust, physics-aware selection policy, we extract scalar invariants that serve as time-series proxies for the system's activity. We clarify that this involves no learnt (ML-based) module. These range from enstrophy flux in 2D or 3D Navier-Stokes equations (Doering & Gibbon, 1995) and the magnitude of outgoing gravitational radiation given by the Weyl scalar $\Psi_4$ (Teukolsky, 1973) or Hamiltonian constraints (Shibata & Nakamura, 1995), to scalar flux fields for gyrokinetic plasma simulations (Krommes, 2012).

We introduce a suite of adaptive temporal selectors designed as online snapshot selector s for two primary use cases: (i) 2D Kolmogorov flows and (ii) 3D BSSN BBH merger simulations. The architecture employs a regulator-decision maker configuration, where a dynamical regulator (regulator) and a physics-aware adaptive threshold (decision maker) operate in feedback to dynamically update the sliding window size $W$ as the simulation streams. This architecture is highly flexible and generalizes across several large-scale scientific computing domains by customizing the decision maker component to monitor specific physical invariants and the subsequent extraction of salient snapshots.

## A.1. 2D Kolmogorov flows

---

**Algorithm 1** 2D Kolmogorov flows in-situ ATS

---

1: **Input:** Trajectory (snapshot) $\mathcal{X} = \{x_t\}_{t=1}^T$, Enstrophy weights $\{\mathcal{E}_t\}_{t=1}^T$, Queue size $Q$, Threshold $\tau$
2: **Output:** Set of selected indices $\mathcal{S}$
3: **Initialize:** $\mathcal{S} \leftarrow \{0\}$, $s \leftarrow 0$ {Start index}
4: $\mathcal{Q} \leftarrow$ empty deque with maxlen $Q$
5: $L \leftarrow 1$ {Last index pointer}
6: **while** $s < T - 1$ **do**
7:     Compute enstrophy difference: $\delta_t = |\mathcal{E}_t - \mathcal{E}_{t-1}|$ for $t \in \mathcal{Q}$
8:     **if** $|\mathcal{Q}| = Q$ or $L = T - 1$ **then**
9:         $\bar{\delta} \leftarrow \frac{1}{|\mathcal{Q}|} \sum_{t \in \mathcal{Q}} \delta_t$ {Mean local enstrophy}
10:        $\delta_{max} \leftarrow \max_{t \in \mathcal{Q}} \delta_t$ {Peak transient activity}
11:        $\eta \leftarrow \sqrt{\delta_{max}/(\bar{\delta} + \epsilon)}$ {Stability-weighted gating factor}
12:        $i^* \leftarrow s + 1$
13:        $\Delta e_0 \leftarrow |\mathcal{E}_{s+1} - \mathcal{E}_s|$
14:        **for** $i \in \mathcal{Q}$ **do**
15:           $\Delta\mathcal{E}_i \leftarrow |\mathcal{E}_i - \mathcal{E}_s|$ {Physics divergence}
16:           $\rho \leftarrow$ pearson$(x_i, x_s)$ {Structural correlation}
17:           **if** $\Delta\mathcal{E}_i/(\Delta\mathcal{E}_0 + \epsilon) \leq \eta$ **and** $\rho \geq \tau$ **then**
18:              $i^* \leftarrow i$
19:           **else**
20:              **break** {Violation of stability or correlation}
21:           **end if**
22:        **end for**
23:        $s \leftarrow i^*$
24:        $\mathcal{S} \leftarrow \mathcal{S} \cup \{i^*\}$
25:        **while** $|\mathcal{Q}| > 0$ **and** $\mathcal{Q}[0] \leq s$ **do**
26:           Pop leftmost element from $\mathcal{Q}$
27:        **end while**
28:     **end if**
29:     **if** $L + 1 < T$ **then**
30:        $L \leftarrow L + 1$
31:        Push $L$ to $\mathcal{Q}$
32:     **end if**
33: **end while**
34: **return** $\mathcal{S}$

---

- **Dynamic Adaptation:** The regulator enforces invariance to non-stationary scale-drift resulting from turbulence variations and is structurally a dynamically adapting, look-forward sliding window buffer. The stability factor $\eta = \sqrt{\delta_{\max}/(\bar{\delta} + \epsilon)}$ functions as a non-linear physics-based regulator. In regimes where the enstrophy flux is highly volatile (high $\delta_{\max}$ relative to $\bar{\delta}$), $\eta$ increases. This allows the algorithm to extend the selection window during stable, quasi-equilibrium regions. During "physically violent" transitions it opts for an higher subsampling where the enstrophy flux exceeds local averages.

- **Spatiotemporal Cross-Correlation Metric:** In the case of turbulent flows, vorticity fields often manifest as high-frequency spatial components at the pixel level. Thus, the temporal selector risks skipping these frames due to exhibiting marginal temporal variance between successive snapshots. To prevent the loss of these multiscale features, we introduce a *spatiotemporal cross-correlation metric* that couples the outer sampling loop with the inner neural field (NF) parametrization. By enforcing a structural correlation threshold, $c > \text{corr}_{\text{threshold}}$, the feedback mechanism ensures that spatially complex snapshots are also prioritized for sampling even when temporal changes within the sliding window W are minimal. This coupling ensures that spatially dense features which are temporally persistent are accurately preserved in the latent representation.

- **Dual-Constraint Gating:** By coupling the physical criterion ($\Delta \mathcal{E}_i / \Delta \mathcal{E}_0 \leq \eta$) with the Statistical Criterion ($\rho \geq \tau$) which is the cross-correlation metric defined above, the selector ensures temporal sparsity without sacrificing physical fidelity. Snapshots are only bypassed if the state remains both physically bounded on the enstrophy submanifold and structurally correlated with the preceding reference frame.

- **In-Situ Efficiency:** The implementation maintains an asymptotic time complexity of $\mathcal{O}(T)$ and a constant memory overhead of $\mathcal{O}(Q)$. This does present some operational constraints of in-situ neural compression (NC) in large-scale volumetric simulations, if the mesh-cells are in billions or trillions.

## A.2. 3D BSSN BBH Merger Simulations

- **Robust Surge Detection via Median-Baseline Tracking**: The regulator utilizes a rolling history buffer $\mathcal{H}$ to maintain a robust estimate of the quiescent field activity using a median filter. By defining the surge threshold as a multiple of the median ($\gamma \cdot \tilde{a}$), the algorithm remains invariant to the low-amplitude numerical noise typical of the pre-merger inspiral phase. This robust baseline allows for the identification of the "surge" regime, where there is a non-linear increase in $||\Psi_4||$ amplitude as the black holes approach merger.

- **Look-ahead Clamping and Mode Switching**: The selector operates in two distinct modes: a quiescent regime (pre and post merger phases) and a surge regime. In the quiescent regime, the algorithm attempts a maximal temporal jump of size $W$. However, it employs a look-ahead verification step that proactively scans the window. If a surge is detected within the look-ahead horizon, the algorithm "clamps" the jump at the surge boundary, effectively downshifting to unit-stride sampling ($t + 1$) before the transient occurs. This ensures that the high-frequency dynamics of the merger are captured with zero latency

- **Hysteresis-Aware History Updates**: To prevent the baseline $\tilde{a}$ from being corrupted by the extreme transients of the merger event, the algorithm implements a gated history update. The history deque $\mathcal{H}$ is only updated when the system is stable ($C_{\text{surge}} = 0$) or after a sustained "new" steady state is established ($C_{\text{surge}} > P$). This patience-based gating ensures that the regulator does not "normalize" the merger signal, and maintains a threshold from the premerger phase. This maintains a high sensitivity to salient physics throughout the entire trajectory.

---

**Algorithm 2** 3D BSSN BBH merger in-situ ATS

---

1: **Input:** Activity series $\mathcal{A} = \{||\Psi_4||_0, \ldots, ||\Psi_4||_{T-1}\}$, window size $W$, threshold factor $\gamma$, patience P, history maxlen $h$, warmup steps $w$
2: **Output:** Selected indices $\mathcal{S}$
3: **Initialize:** $\mathcal{S} \leftarrow \{0\}, t \leftarrow 0, C_{\text{surge}} \leftarrow 0, \mathcal{H} \leftarrow \text{deque}(\text{maxlen} = h)$
4: **while** $t < T - 1$ **do**
5:    **if** $|\mathcal{H}| < w$ **then**
6:       {Initial profiling of the quiescent field}
7:       $t_{\text{next}} \leftarrow t + 1$
8:       $C_{\text{surge}} \leftarrow 0$
9:    **else**
10:      $\tilde{a} \leftarrow \text{median}(\mathcal{H})$ {Robust baseline estimation}
11:      $a_{t+1} \leftarrow \mathcal{A}[t+1]$
12:      **if** $a_{t+1} > \gamma \cdot \tilde{a}$ **then**
13:        {Transient $||\Psi_4||$ surge detected}
14:        $t_{\text{next}} \leftarrow t + 1$ {Force dense (unit) sampling}
15:        $C_{\text{surge}} \leftarrow C_{\text{surge}} + 1$
16:      **else**
17:        {Baseline regime: attempt temporal jump}
18:        $C_{\text{surge}} \leftarrow 0$
19:        $t_{\text{cand}} \leftarrow \min(t + W, T - 1)$
20:        $t_{\text{next}} \leftarrow t_{\text{cand}}$
21:        **for** $k = t + 1$ **to** $t_{\text{cand}}$ **do**
22:           {Look-ahead for merger onset}
23:           **if** $\mathcal{A}[k] > \gamma \cdot \tilde{a}$ **then**
24:             $t_{\text{next}} \leftarrow k$ {Clamp jump at surge boundary}
25:             **break**
26:           **end if**
27:        **end for**
28:      **end if**
29:    **end if**
30:    $t \leftarrow t_{\text{next}}$
31:    $\mathcal{S} \leftarrow \mathcal{S} \cup \{t\}$
32:    {Update $\mathcal{H}$ only if the system is stable or has entered a new steady state}
33:    **if** $C_{\text{surge}} = 0$ **or** $C_{\text{surge}} > P$ **then**
34:      $\mathcal{H}.\text{append}(\mathcal{A}[t])$
35:      **if** $C_{\text{surge}} > P$ **then**
36:        $C_{\text{surge}} \leftarrow 0$
37:      **end if**
38:    **end if**
39: **end while**
40: **return** $\mathcal{S}$

---

---

**Algorithm 3 Adaptive Neural Temporal In-situ Compressor (ANTIC)**

---

**Require:** Base NF $\mathbf{W}_0$; Physics metrics $\{\phi_t\}$; Window size $W$; Simulation duration $T$, simulation snapshot $u_t$
**Ensure:** Compressed updates $\{\Delta\mathbf{W}\}$
1: Initialize $t \leftarrow 0, \mathcal{S} \leftarrow \{0\}$
2: **while** $t < T$ **do**
3:     {I. Physics-Aware Temporal Selection (PATS) using Algos such as (1, 2)}
4:     $\Delta t \leftarrow \text{PATS}(\{\phi_k\}_{k=t}^{t+W}, W)$ {Stride $\in \{1, \dots, W\}$ based on saliency}
5:     $t \leftarrow t + \Delta t$
6:     $\mathcal{S} \leftarrow \mathcal{S} \cup \{t\}$
7:     {II. Neural Compression (NC) Module}
8:     Fetch snapshot $u^t$ from simulation
9:     **if** Mode is FULL FT **then**
10:         $\mathbf{W}_t \leftarrow \text{Optimize}(\mathbf{W}_{t-\Delta t}, u^t)$
11:     **else if** Mode is LoRA **then**
12:         $\mathbf{B}, \mathbf{A} \leftarrow \text{OptimizeAdapters}(\mathbf{W}_{t-\Delta t}^r, u^t)$ {Rank-$r$ update}
13:         $\Delta\mathbf{W}_t \leftarrow \mathbf{BA}$
14:         $\mathbf{W}_t \leftarrow \mathbf{W}_{t-\Delta t} + \Delta\mathbf{W}_t$ {Merge update into base weights}
15:         **Reset** adapters $\mathbf{B}, \mathbf{A} \leftarrow \mathbf{0}$ {Prepare for next selected snapshot}
16:     **end if**
17:     Store/Transmit compressed update $\Delta\mathbf{W}_t$ on persistent storage
18: **end while**
19: **return** Selection set $\mathcal{S}$ and compressed updates

---

# B. Miscellaneous Experimental Results

## B.1. (2D) Kolmogorov flows

### B.1.1. EXPERIMENTAL SETUP, DATA PREPARATION AND NF TRAINING SPECIFICS

Incompressible fluid dynamics are governed by the Navier–Stokes equations, which represent the conservation of momentum and mass for a Newtonian fluid:

$$\frac{\partial \mathbf{u}}{\partial t} + (\mathbf{u} \cdot \nabla)\mathbf{u} = -\frac{1}{\rho}\nabla p + \nu\nabla^2\mathbf{u} + \mathbf{f}, \tag{2}$$

$$\nabla \cdot \mathbf{u} = 0, \tag{3}$$

where $\mathbf{u}(\mathbf{x}, t)$ is the velocity field, $p$ is the scalar pressure field, and $\mathbf{f}$ denotes external body forces. The kinematic viscosity $\nu$ is related to the dimensionless Reynolds number by $Re = UL/\nu$, where $U$ and $L$ are characteristic velocity and length scales, respectively. In this formulation, pressure serves as a Lagrange multiplier that constrains the velocity field to the solenoidal (divergence-free) manifold. At high Reynolds numbers ($Re \gg 1$), the flow enters a turbulent regime characterized by a nonlinear energy cascade and high-frequency spatiotemporal fluctuations. From a neural compression perspective, these regimes are particularly demanding; the emergence of fine-scale turbulent eddies increases the spectral complexity of the data, requiring the neural field to resolve a wide range of spatial frequencies while capturing rapid temporal transients that limit the effective window size of standard temporal regulators. Specifically, we choose decaying turbulence, which refers to a canonical flow regime where an initial high-energy, turbulent velocity field evolves in the absence of external forcing ($\mathbf{f} = 0$). Unlike forced turbulence, which reaches a statistical steady state, decaying turbulence is a transient phenomenon where kinetic energy is progressively dissipated into heat through viscous effects at the smallest scales.

In order to generate the turbulent flow simulations, we use (Dresdner et al., 2022) which primarily uses a semi-implicit Crank-Nicholson Runge-Kutta order 4 solver to simulate decaying Kolmogorov turbulence. The system is evolved on a $[0, 2\pi]^2$ periodic domain with a spatial resolution of $2048 \times 2048$. We perform $1,000$ time steps to a terminal time $T = 25.0\,\text{s}$ ($\eta = 10^{-3}$, $v_{\text{init}} = 5.0$, $Re = 1000$). The resulting scalar vorticity dataset for the entire trajectory being $16.8\,\text{GiB}$ and providing a dense multiscale benchmark where high-frequency spatial features and fast dynamical motion are accompanied by low-frequency and slowly evolving regions.

**Neural field training specifics.** We set the FFM embedding frequency especially playing a pivotal role in parametrizing 2D

Kolmogorov flows snapshots containing high-frequency components due to high vorticity field magnitude reflecting at the pixel-level. Thus, we opt for a Fourier embedding of dim 256 with a embedding frequency of 7.0.

## B.2. (3D) BSSN evolution

### B.2.1. EXPERIMENTAL SETUP, DATA PREPARATION AND NF TRAINING SPECIFICS

# C. Numerical Relativity and the BSSN Formulation

The numerical solution of Einstein's field equations is central to characterizing astrophysical scenarios where gravity is strong and highly dynamical. Within the numerical relativity 3+1 decomposition, the 4D spacetime metric $g_{\mu\nu}$ is partitioned via the ADM split:

$$ds^2 = -\alpha^2 dt^2 + \gamma_{ij}(dx^i + \beta^i dt)(dx^j + \beta^j dt), \tag{4}$$

where $\alpha$ represents the lapse function, $\beta^i$ is the shift vector, and $\gamma_{ij}$ denotes the induced spatial metric on Cauchy hypersurfaces. The corresponding four-metric is expressed as:

$$g_{\alpha\beta} = \begin{pmatrix} -\alpha^2 + \beta_i\beta^i & \beta_i \\ \beta_i & \gamma_{ij} \end{pmatrix}. \tag{5}$$

The Baumgarte-Shapiro-Shibata-Nakamura (BSSN) formulation improves upon the standard ADM formalism by introducing conformal transformations that enhance numerical stability and constraint preservation. BSSN evolves a set of conformal variables, including the conformal metric $\tilde{\gamma}_{ij} = W^2\gamma_{ij}$ (where $W$ is the conformal factor) and the trace-free extrinsic curvature $\tilde{A}_{ij}$. This framework is particularly robust for simulating binary black hole (BBH) mergers using the "moving punctures" approach, where singularities are handled via a conformal factor that vanishes at the puncture locations.

## C.1. Extraction of Gravitational Radiation: The Weyl Scalar $\Psi_4$

To quantify the gravitational radiation generated during the BBH merger, we compute the Newman-Penrose scalar $\Psi_4$. This scalar represents the outgoing transverse-traceless component of the gravitational field at the extraction boundary. In the 3+1 formalism, $\Psi_4$ is reconstructed from the electric ($E_{ij}$) and magnetic ($B_{ij}$) parts of the Weyl tensor:

$$E_{ij} = R_{ij} + KK_{ij} - K_{ia}K_j^a, \quad B_{ij} = \epsilon_{iak}D^a K_j^k, \tag{6}$$

where $R_{ij}$ is the 3-Ricci tensor and $D_i$ is the spatial covariant derivative. Given a complex null tetrad $\{l, n, m, \bar{m}\}$, the Weyl scalar is projected as:

$$\Psi_4 = -C_{\alpha\beta\gamma\delta} n^\alpha \bar{m}^\beta n^\gamma \bar{m}^\delta. \tag{7}$$

Using the decomposition into electric and magnetic components, this projection simplifies to:

$$\Psi_4 = (E_{ij} - iB_{ij})\bar{m}^i\bar{m}^j. \tag{8}$$

In our PATS regulator for the 3D BBH merger simulation, the magnitude $||\Psi_4(t, \mathbf{r})||$ extracted at a radii $\mathbf{r}$, serves as the primary physics-aware metric. It provides a direct measure of the radiative activity in the spacetime, allowing the selector to identify the high-frequency transients associated with the merger and ringdown phases.

## C.2. Experimental Configuration

The simulation utilizes a Cartesian grid of size $213^3$ within a domain range of $[-15M, 15M]$. We initialize a non-spinning BBH system with equal bare masses $M_1 = M_2 = 0.483$ (total mass $M \approx 1$). The punctures are positioned at $y = \pm 3.257$ with initial momenta $P_x = \mp 0.133$. The evolution spans $T = 250M$ over 5,966 steps using an implicit backward Euler integrator. For neural compression, we exclude gauge-dependent shift vectors but retain the lapse function for visualization, resulting in a training set of 18 tensor components per selected snapshot.

# D. Extended Pareto-Frontier over Model Size Sweeps

Here, we present a pareto-frontier over different model size sweeps ($32 \times 2 \to 512 \times 8$) for the varied training schemes are shown. Here too we compare CFT (continual FT) and the LoRA $\{8, 16, 32, 64\}$. This is an extended version of the pareto-frontier for the model utilized, i.e. $256 \times 6$ (see Figs. 4a, 4b) presented in the main paper, which primarily used full finetuning as the reference baseline as compared to other training schemes. As evident from the pareto in Figs. (5a, 5b) that continual fine-tuning (CFT) already reaches comparable accuracy compared to cold-start trainings, while offering extreme compression per snapshot for high-dimensional multi-scale BSSN simulations requiring high-resolution explicit grids.

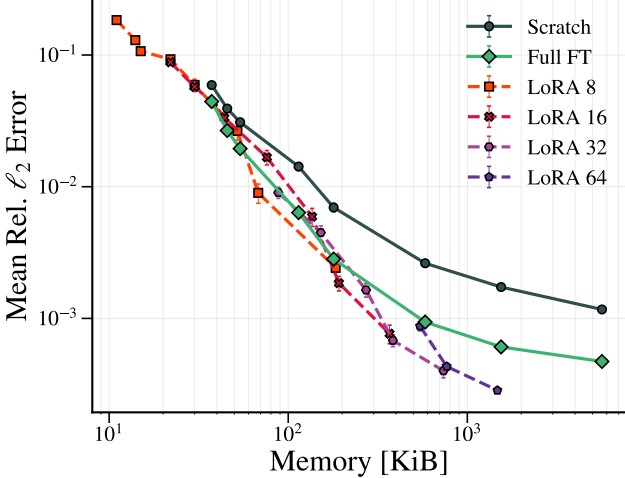
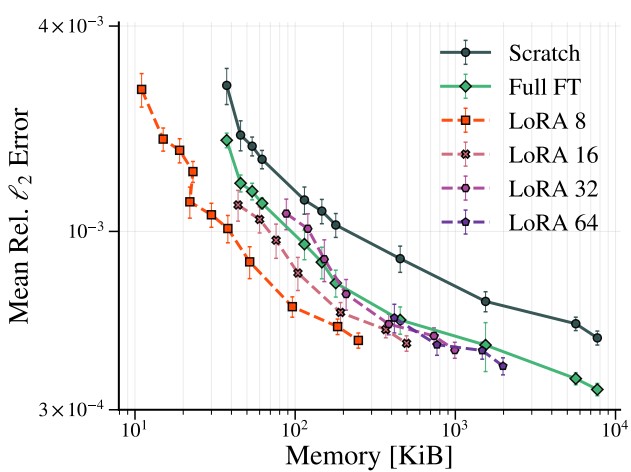

*(a)* **2D Kolmogorov flow extended pareto-front for spatial compression.** Averaged relative $\ell_2$ error over 5 subsequent frames in highly-turbulent window as a function of compression rate (KiB) for different model sizes. We compare three training paradigms: (i) compressing each snapshot from scratch, (ii) continual fine-Tuning (FT), and (iii) CFT via LoRA ($r \in [8, 64]$). LoRA ($r > 32$) attains a comparable error to FT while significantly improving compression rate. The LoRA trendlines are truncated to satisfy the efficiency constraint $r \leq d/2$, where $d$ is the hidden dimension. We omit configurations where the rank exceeds this threshold—such as $r \geq 32$ for the $32 \times L$ model suite

*(b)* **3D BBH merger pareto-front for spatial compression.** Averaged relative $\ell_2$ error over 5 subsequent frames in the near merger regime as a function of parameter memory (KiB). We compare three training paradigms: (i) compressing each snapshot from scratch, (ii) continual fine-Tuning (FT), and (iii) CFT via LoRA ($r \in [8, 64]$). LoRA-edited NFs (ranks $r \geq 8$) attain a comparable error to FT while significantly improving compression rate. The LoRA trendlines are truncated to satisfy the efficiency constraint $r \leq d/2$, where $d$ is the hidden dimension. We omit configurations where the rank exceeds this threshold—such as $r \geq 32$ for the $32 \times L$ model suite

# E. Benchmarking Continual Learning Against Cold-Start Methodologies

Multi-scale physics simulations exhibit dynamical regimes that can be broadly categorized as fast-varying (high turbulence regions, phase transitions, merger regimes with non-trivial changes in spacetime geometry) and slow-varying (quasi-static evolution). Outside of sharp transition regions, the state of subsequent snapshots separated by $\Delta t$ does not vary drastically, i.e., $\Delta u := \|u(\mathbf{x}, t + \Delta t) - u(\mathbf{x}, t)\|$ remains small. This property enables the use of continual learning, since the light weight updates $\mathbf{W}_{t+\Delta t} - \mathbf{W}_t$ between subsequent neural field parametrizations need only encode these perturbative changes.

This substantially reduces the training wall-clock time for the CFT-based methodology. Convergence to the same target accuracy of $3 \times 10^{-8}$ is achieved in only 3 epochs, compared to 35 epochs under the cold-start strategy, representing a speedup exceeding $7\times$. This improvement in training overhead makes CFT well-suited for in-situ scenarios, where training times must remain commensurate with solver runtimes in an asynchronous pipeline. Furthermore, CFT is directly compatible with LoRA-based neural field compression (Truong et al., 2025), enabling extreme spatial compression alongside significantly faster convergence (exceeding $10\times$) relative to cold-start methodologies. These are reported in Table 4. Additional, we report the benchmark for these experiments conducted on both: a single NVIDIA H200 (144 GiB, driver version $- 590.48.01$) and also on NVIDIA Blackwell B300 SXM6 AC GPU (275 GiB, driver version $- 590.48.01$), which is $4 - 5\times$ faster than H200s for the multi-layer perceptron trainings, since they are optimized for AI workfloads utilizing the Tensor cores.

*Table 4.* **Hardware Benchmark: Continual Fine-Tuning (CFT) vs. Solver vs. Cold-Start.** Comparison of wall-clock convergence times across **NVIDIA H200** and **B300 (Blackwell)** architectures for the 3D BSSN system. CFT updates perturbative changes between snapshots, achieving an order-of-magnitude speedup over cold-Starts while maintaining a comparable MSE of $3 \times 10^{-8}$. The BSSN solver is a bit slower on B300s due to skinny batched **GEMM** — millions of tiny einsum-based contraction operations that are lowered to `dot_general` on cuBLAS for which below the Tensor Core tile threshold, rendering the solver memory-bandwidth-bound rather than compute-bound.

| HARDWARE | METHOD | WALLCLOCK (S) | EPOCHS | MSE |
|---|---|---|---|---|
| **H200** | BSSN SOLVER (GPU) | $\sim 0.25 - 0.27$ | – | – |
| | COLD START | $\sim 183 - 187$ | 35 | $\sim 3 \times 10^{-8}$ |
| | **CFT (OURS)** | $\sim \mathbf{21 - 24}$ | **3** | $\sim \mathbf{3 \times 10^{-8}}$ |
| **B300** | BSSN SOLVER (GPU) | $\sim 0.27 - 0.44$ | – | – |
| | COLD START | $\sim 52 - 54$ | 35 | $\sim 3 \times 10^{-8}$ |
| | **CFT (OURS)** | $\sim \mathbf{4.5 - 4.7}$ | **3** | $\sim \mathbf{3 \times 10^{-8}}$ |

Thus, CFT-based continual learning scheme exploits inter-snapshot continuity to reduce per-snapshot encoding cost from $\mathcal{O}(T_{\text{cold}})$ to $\mathcal{O}(T_{\text{CFT}}) \ll T_{\text{cold}}$.

## F. Comparison against Other Classical Compressors in Scientific Computing

We benchmark against four compressors representing the dominant paradigms in scientific lossy and lossless compression: SZ3 (Liang et al., 2021), ZFP (Lindstrom, 2014), MGARD (Gong et al., 2023a), and Blosc-2 (Blosc Development Team, 2009-2025). **SZ3.** (Liang et al., 2021) is a modular prediction-based lossy compressor for floating-point data. It predicts field values via Lorenzo or regression predictors, quantizes the residuals, and applies Huffman and Lempel-Ziv entropy coding. Error bounds are enforced strictly in the absolute or relative $\ell^{\infty}$ sense. SZ3 achieves high compression ratios on smooth fields, but prediction residuals grow large in turbulent or discontinuous regimes, limiting its effectiveness at tight accuracy tolerances.

**ZFP.** (Lindstrom, 2014) partitions d-dimensional arrays into $4^d$ 4d blocks and applies a near-orthogonal integer transform within each block, followed by embedded bit-plane encoding. It supports fixed-rate, fixed-precision, and fixed-accuracy modes. Its block-local structure limits exploitation of long-range correlations, and compression quality degrades in fields with sharp multi-scale features, where energy leaks across bit planes within discontinuous blocks.

**MGARD.** (Gong et al., 2023a) is grounded in multigrid theory, decomposing fields onto a hierarchy of nested grids and encoding the multilevel coefficients with quantization and entropy coding. It provides guaranteed error bounds in the $\ell^2$ or $\ell^{\infty}$ norm, and supports error control on derived quantities of interest. It is most effective on globally smooth fields where coarse-scale grid levels capture the dominant energy content and is GPU supported.

**Blosc-2.** (Blosc Development Team, 2009-2025) is a lossless meta-compressor that applies byte- or bit-shuffling filters to floating-point data prior to entropy coding via Zstd or LZ4. As a lossless method, it preserves the data exactly, achieving typical compression ratios of $2\times$-$5\times$ on scientific fields, and serves as an upper bound on compression achievable under zero information loss.

**Positioning against Spatial Neural Compression.** These four compressors share a common limitation from the perspective of in-situ scientific workflows: they operate on individual snapshots independently, without exploiting the *temporal continuity* of the simulation trajectory. SZ3, ZFP, and MGARD compress each snapshot in isolation, discarding the perturbative structure $\Delta u := \|u(\mathbf{x}, t + \Delta t) - u(\mathbf{x}, t)\|$. that characterizes neighboring snapshots that do not change drastically. Furthermore, their compressed representations are not *queryable* — reconstructing a physical quantity at an arbitrary spatial location requires full decompression of the relevant block or subgrid. ANTiC addresses both limitations simultaneously: the neural field parametrization provides a continuous, AD-based differentiable implicit representation of the field that supports point-wise queries without full decompression. The numbers for all these compressors against our CFT-enhanced neural compression is reported in Table 5

*Table 5.* **Performance Benchmarks of Traditional vs. Neural Compression**. Comparison of Compression Ratio (CR), Wall-clock Time, and Relative $\ell_2$ Error across 2D Kolmogorov flows vorticity field ($2048^2$) in the turbulent regime and large-scale 3D BSSN evolved variables ($\sim 0.7$ GiB/snapshot, ($213^3$, 18) shaped array) in the premerger phase. While traditional compressors (SZ3, ZFP, MGARD and Blosc-2) maintain low latency for small-scale 2D data, our NC with (CFT) particularly the LoRA variant achieves an unprecedented $3745\times$ compression ratio for tensor-valued BSSN variables. This represents a two-order-of-magnitude improvement over state-of-the-art error-bounded compressors (SZ3) while maintaining comparable reconstruction fidelity ($\sim 10^{-4}$), effectively enabling the storage of high-resolution multi-scal simulations with a minimal memory footprint on the persistent storages. The neural field compression numbers reported are conducted on the B300 GPU.

| COMPRESSOR | 2D KOLMOGOROV (16.8 MiB/SNAPSHOT) | | | 3D BSSN (0.7 GiB/SNAPSHOT) | | |
|---|---|---|---|---|---|---|
| | CR ($\uparrow$) | Time ($\downarrow$) | Rel $\ell_2$ ($\downarrow$) | CR ($\uparrow$) | Time ($\downarrow$) | Rel $\ell_2$ ($\downarrow$) |
| ZFP (TOL=1E−03) | $8\times$ | **0.06** (s) | **1.02e-4** | $4\times$ | **2.82** (s) | **1.27e-4** |
| MGARD [CPU] (ABS=1E−03) | $19\times$ | 0.37 (s) | 4.02e-4 | $5\times$ | 38.67 (s) | 7.97e-4 |
| SZ3 (ABS=1E−03) | **154**$\times$ | 0.07 (s) | 8.31e-4 | $16\times$ | 3.42 (s) | 1.02e-4 |
| BLOSC−2 ZSTD (LOSSLESS) | $1.7\times$ | 1.38 (s) | 0.0e+00 | $1.3\times$ | 54.48 (s) | 0.0e+00 |
| BLOSC−2 ZSTD (LOSSY TRUNC=10) | $5\times$ | 1.03 (s) | 4.23e-4 | $2.8\times$ | 37.82 (s) | 3.93e-4 |
| NEURAL COMPRESSION FT (CFT) | $12\times$ | $\sim 13$ (s) | 6.14e-4 | $471\times$ | $\sim 5$ (s) | 4.60e-4 |
| NEURAL COMPRESSION LoRA (CFT) | $47\times$ [r: 32] | $\sim 18$ (s) | 6.82e-4 | **3745**$\times$ [r: 16] | $\sim 8$ (s) | 5.82e-4 |

## G. Global Reconstruction Quality and Physics Preservation

A critical requirement of ANTIC is the preservation of temporal coherence and high-fidelity reconstruction of physical observables from decompressed neural field parametrizations, including sharp transient features that are particularly sensitive to compression artifacts. Our CFT-based continual learning scheme, stabilized via LayerNorm and weight decay, demonstrates robustness over long-horizon trajectories spanning full simulation runs, maintaining a per-snapshot training MSE of $10^{-7}$-$10^{-9}$ throughout. We evaluate the global reconstruction quality of physically meaningful derived quantities obtained from ANTIC across two distinct simulation regimes: (i) the normalized enstrophy flux for 2D Kolmogorov flows (Subsec. G.1), and (ii) constraint violation diagnostics and geometric quantities for 3D binary black hole merger simulations evolved under the BSSN framework (Subsec. G.2).

### G.1. 2D Kolmogorov flows Enstrophy Flux Reconstruction

For the 2D Navier-Stokes use case, we assess reconstruction fidelity through the normalized enstrophy flux, a scalar diagnostic that directly characterizes the turbulent cascade structure of the flow. Enstrophy, defined as $\mathcal{E}(t) = \frac{1}{2} \int_{\Omega \subset \mathbb{R}^2} ||\omega(x, y, t)||^2 dA$, where $\omega$ is the vorticity field, quantifying the rotational energy content of the flow and governs the forward cascade of energy to small scales in 2D turbulence. The enstrophy flux measures the rate at which enstrophy is transferred across spatial scales, and is thus a stringent probe of whether the compressed representation faithfully preserves the multi-scale vortical structure of the flow — including fine-scale vorticity filaments and the onset of turbulent intermittency. Crucially, errors in the vorticity field are amplified in the enstrophy flux relative to the velocity field, making it a more sensitive diagnostic of compression fidelity than pointwise field metrics alone. We evaluate this quantity over the most dynamically active phase of the simulation, $\tau \in [0, 10]$ (timesteps $[0, 400]$), during which the flow transitions through its peak turbulent regime and enstrophy production is maximal.

*Table 6.* **Reconstruction Fidelity Metrics.** Error analysis for Enstrophy Flux (scalar) and the Vorticity Field (2D grid). The low maximum relative $L_2$ error (0.37%) demonstrates that PATS successfully captures the most physically informative transients in the Kolmogorov flow.

| QUANTITY | METRIC | VALUE |
|---|---|---|
| ENSTROPHY FLUX | MEAN ABSOLUTE ERROR | $1.830 \times 10^{-4}$ |
| | MAX ABSOLUTE ERROR | $2.870 \times 10^{-4}$ |
| VORTICITY FIELD | MEAN RELATIVE $\ell_2$ ERROR | $1.587 \times 10^{-3}$ |
| | MAX RELATIVE $\ell_2$ ERROR | $3.669 \times 10^{-3}$ |

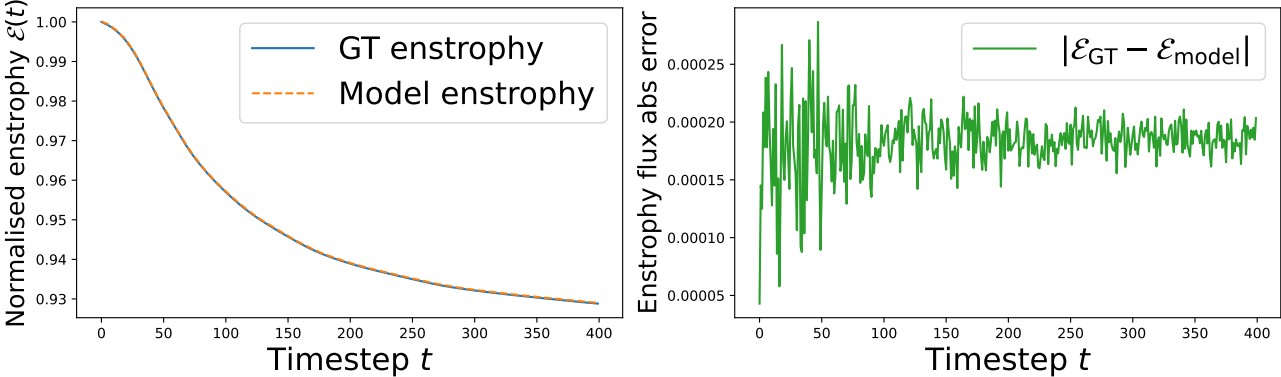

Figure 6. **Enstrophy flux reconstruction for 2D Kolmogorov flow.** (*Left*) Enstrophy flux extracted from ground truth snapshots (solid blue) and ANTIC reconstructions (dashed orange) over the peak turbulent phase $\tau \in [0, 15]$ (timesteps $[0, 400]$). ANTIC faithfully reproduces the temporal evolution of the enstrophy flux, including sharp transient features associated with turbulent intermittency. (*Right*) Pointwise absolute error over the same interval, remaining uniformly small with no systematic drift or error accumulation, confirming the temporal stability of the CFT-based continual learning scheme over long-horizon trajectories.

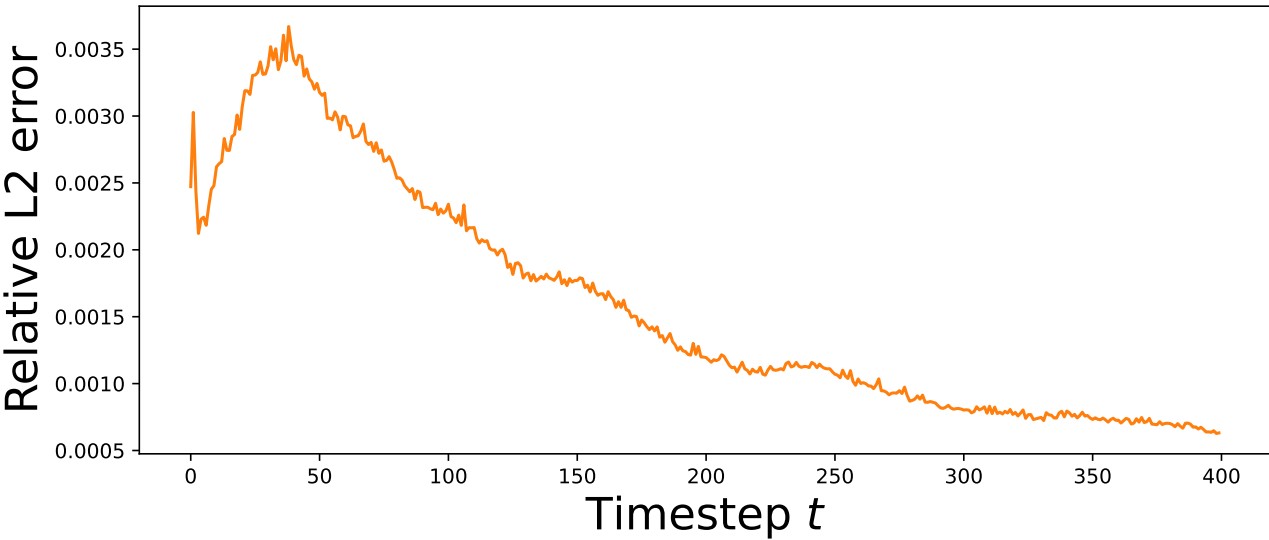

Figure 7. **CFT reconstruction fidelity over long-horizon 2D Kolmogorov flow trajectories.** Relative $\ell_2$ error of ANTIC reconstructions plotted against timestep for the full 2D Kolmogorov flow simulation. The trajectory spans two dynamically distinct regimes: a peak turbulence phase over timesteps $[0, 100]$, characterized by rapid vorticity production, strong enstrophy cascade, and sharp small-scale features that place maximal demand on the neural field parametrization, followed by a gradual decay phase over timesteps $[100, 400]$ as the flow relaxes towards a quasi-steady turbulent state. Despite the elevated reconstruction difficulty during the high-turbulence regime — where inter-snapshot variability $\Delta u := \|u(\mathbf{x}, t + \Delta t) - u(\mathbf{x}, t)\|$ is largest — the CFT-based continual learning scheme maintains a stable relative $\ell_2$ error $< 4 \times 10^{-3}$ in the highest turbulence region and $< 2 \times 10^{-3}$, with no systematic degradation across the trajectory. This confirms that perturbative weight editing between subsequent neural field parametrizations remains well-conditioned even under large dynamical variability, demonstrating the robustness of ANTIC for in-situ compression over long-horizon simulation runs.

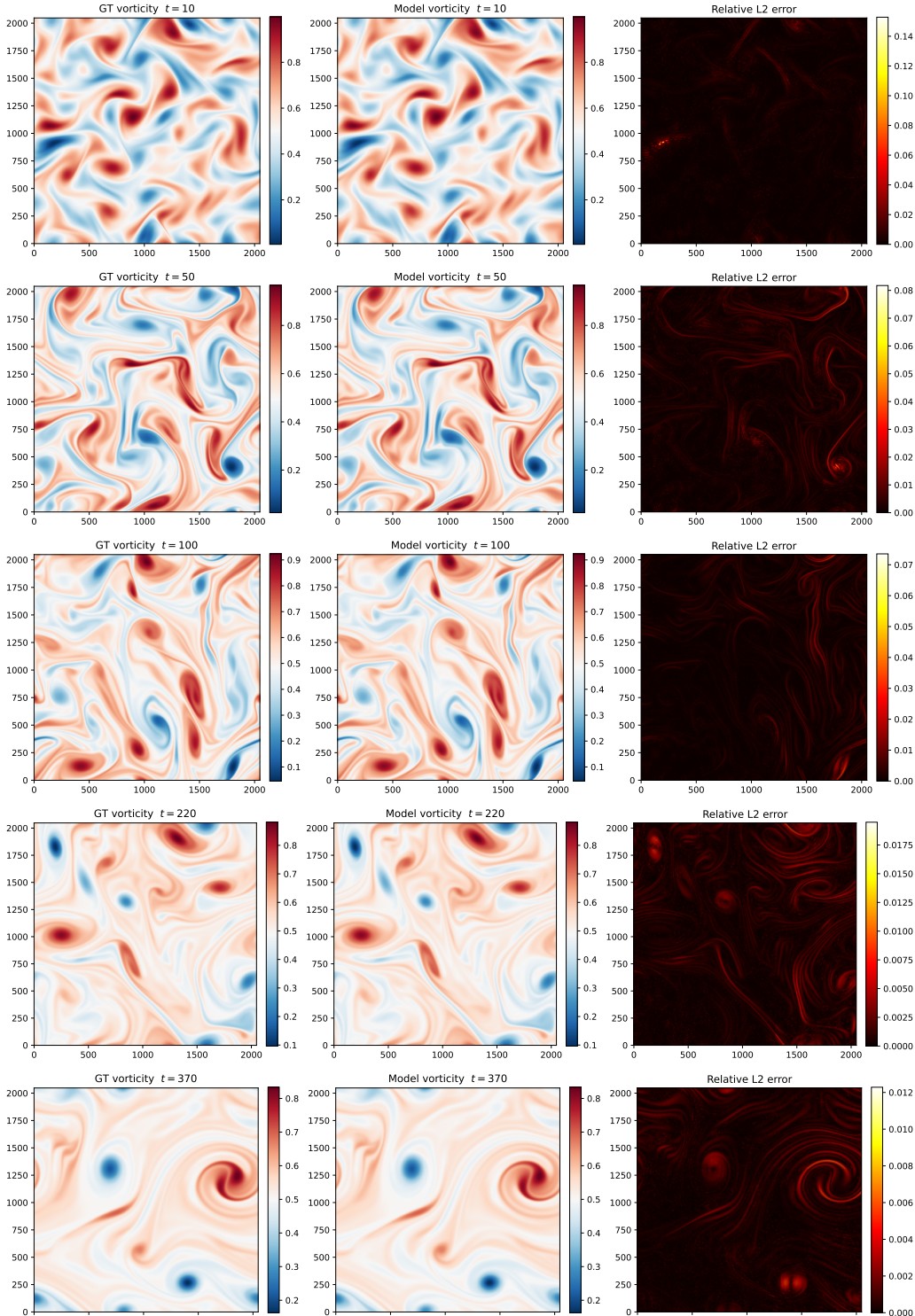

*Figure 8.* **Vorticity field reconstruction quality across dynamical regimes.** Each row corresponds to a representative timestep $t \in \{10, 50, 100, 220, 370\}$, spanning the peak turbulence phase and its subsequent decay, for the 2D Kolmogorov flow simulation at spatial resolution $2048^2$. *First column:* Ground truth vorticity field $\omega(\mathbf{x}, t)$, exhibiting the characteristic fine-scale vorticity filaments, coherent structures, and inter-scale energy transfer that define the turbulent cascade. *Second column:* ANTIC neural field reconstructions at the corresponding timesteps, obtained by querying the continually fine-tuned implicit parametrization. *Third column:* Pointwise relative $\ell_2$ error between the ground truth and reconstructed vorticity fields, confirming that reconstruction fidelity is maintained across both the high-turbulence regime - where vorticity gradients are steepest and small-scale structures are most pronounced — and the quasi-steady decay phase (after timestep $t > 150$), with no systematic spatial concentration of errors around coherent vortical structures.

## G.2. 3D BBH Merger Weyl Scalar Magnitude Reconstruction

The Weyl scalar $\Psi_4$ defined in Eqs. (7, 8) is a coordinate-invariant complex scalar quantity extracted from the Weyl curvature tensor that encodes the outgoing gravitational radiation content of the spacetime. In the wave zone, $\Psi_4$ corresponds directly to the second time derivative of the gravitational wave strain, making it the primary observable for characterizing the amplitude, frequency, and phase evolution of gravitational waves emitted during compact binary mergers.

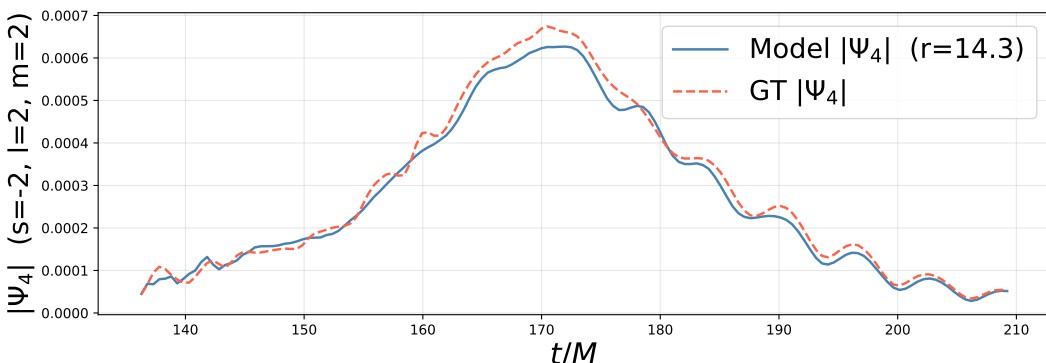

*Figure 9.* **Weyl scalar magnitude reconstruction during the BBH merger phase.** The gravitational wave signal $|\Psi_4(t/M)|$ extracted from ground truth snapshots (solid) and `ANTIC` neural field reconstructions (dashed) at extraction radius $r = 14.3\,M$, over the merger phase $t/M \in [136, 210]$. This interval encompasses the peak gravitational wave emission, ringdown onset, and the most dynamically complex regime of the spacetime evolution, placing maximal demand on the fidelity of the neural field parametrization. `ANTIC` accurately reproduces the amplitude and phase evolution of the Weyl scalar throughout, including the characteristic amplitude peak at merger.

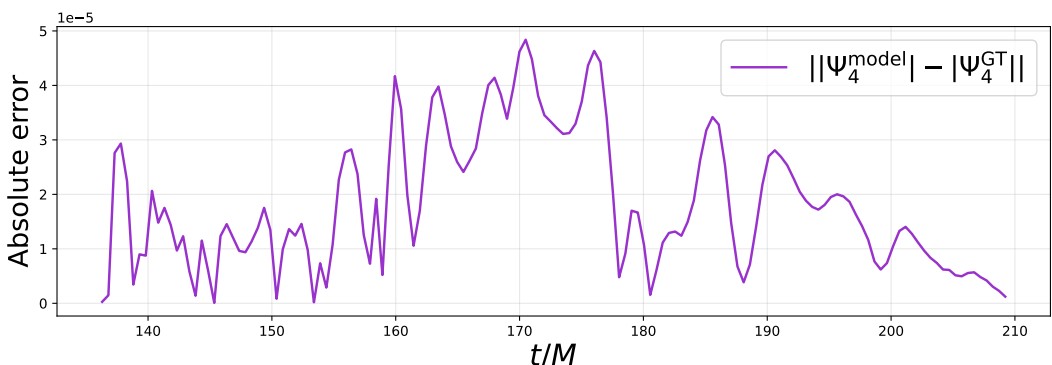

*Figure 10.* **Absolute reconstruction error in Weyl scalar magnitude.** Pointwise absolute error $\left| |\Psi_4^{\mathrm{model}}(t/M)| - |\Psi_4^{\mathrm{GT}}(t/M)| \right|$ at extraction radius $r = 14.3\,M$ over $t/M \in [136, 210]$. Errors remain uniformly small across the full merger phase, with no systematic growth near the amplitude peak, confirming that `ANTIC` preserves the gravitational wave content of the spacetime to high fidelity under the CFT-based continual learning scheme.

*Table 7.* **BSSN Reconstruction Performance.** Error analysis for the Weyl Scalar magnitude $|\Psi_4|$ during the Binary Black Hole (BBH) merger and initial post-merger regime ($t/M \in [136.30, 209.22]$). The results compare 146 snapshots across the peak gravitational wave emission phase.

| CATEGORY | METRIC | NEF (OURS) | GROUND TRUTH |
|---|---|---|---|
| $|\Psi_4|$ RANGE | MINIMUM VALUE | $2.825 \times 10^{-5}$ | $3.381 \times 10^{-5}$ |
| | MAXIMUM VALUE | $6.267 \times 10^{-4}$ | $6.741 \times 10^{-4}$ |

| METRIC | QUANTITY | VALUE |
|---|---|---|
| ERROR | MEAN ABSOLUTE ERROR | $1.835 \times 10^{-5}$ |
| | MAX ABSOLUTE ERROR (AT $t/M = 170.49$) | $4.838 \times 10^{-5}$ |

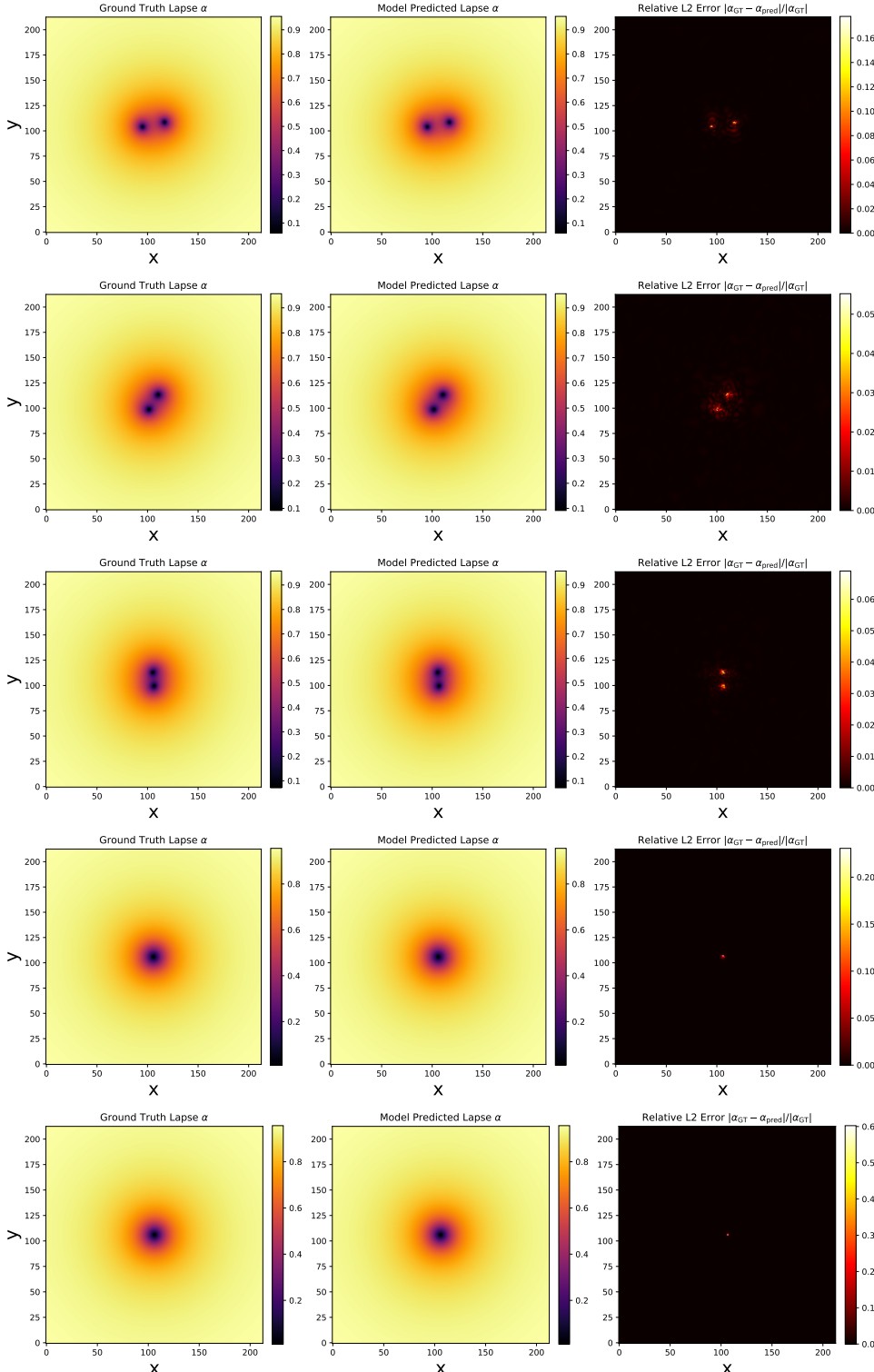

*Figure 11.* **Lapse function reconstruction quality across the BBH merger trajectory.** Each row corresponds to a representative timestep $t/M \in \{136.30, \ldots, 209.22\}$, spanning the inspiral-to-merger transition and post-merger ringdown phase of the 3D BBH simulation at spatial resolution $213^3$. *First column:* Ground truth lapse function $\alpha(\mathbf{x}, t)$ evolved under the BSSN framework (Sec. B.2), exhibiting the characteristic collapse toward zero in the black hole interior regions as the merger proceeds. *Second column:* ANTIC neural field reconstructions at the corresponding timesteps, obtained by querying the continually fine-tuned implicit parametrization. *Third column:* Pointwise relative $\ell_2$ error between the ground truth and reconstructed lapse fields, confirming that ANTIC faithfully captures both the smooth exterior spacetime geometry and the sharp gradients near the apparent horizons, with no systematic spatial concentration of errors in the dynamically critical merger region.

## H. Comparing Temporal Retention of Physics-Agnostic selectors against PATS

The primary function of the PATS submodule is the extraction of salient snapshots from simulation trajectories via in-situ adaptive time-step control, applying fine temporal sampling during fast-evolving regimes and coarse sampling during quasi-static evolution, guided by physics-based scalar diagnostics computed directly from each snapshot. For completeness, we also consider physics-agnostic temporal selectors, including momentum-aware selectors (Jha et al., 2025; Zhang & Gao, 2025) developed for 3D scene reconstruction and visual computing, and Kullback-Leibler (KL) divergence based selectors (Yamaoka et al., 2019), which require no in-situ scalar signal extraction and operate independently of the underlying physical dynamics. We demonstrate that while such selectors are well-suited to vision-centric tasks, they fail to identify dynamically critical transients in PDE simulation trajectories. We evaluate the following temporal selectors: **Momentum-aware** (Jha et al., 2025), and four **entropy-based** variants: (i) *Jensen-Shannon divergence (JSD)*, (ii) *residual differential entropy (Res)*, (iii) *spectral entropy (Spectral)*, and (iv) *normalized mutual information (MI)*. All evaluations are performed on the 2D Kolmogorov flow use case at resolutions $1024^2$ and $2048^2$, which is sufficient to expose the failure modes of physics-agnostic selectors relative to PATS. Temporal retention plots are reported for the *JSD* selector: Figs. (12a, 12b), *Spectral* selector: Figs. (13a, 13b), *Residual* selector: Figs. (14a, 14b), *MI* selector: Figs. (15a, 15b), and *Momentum*-aware selector: Figs. (16a, 16b). Each is compared against PATS, which uses the enstrophy flux as its decision signal, retaining snapshots containing high-turbulence transients at fine temporal resolution while coarsening the sampling rate during equilibration: Figs. (17a, 17b). Quantitative temporal retention statistics across additional spatial resolutions are reported in Table 8, further corroborating the necessity of physics-based adaptive sampling over physics-agnostic alternatives.

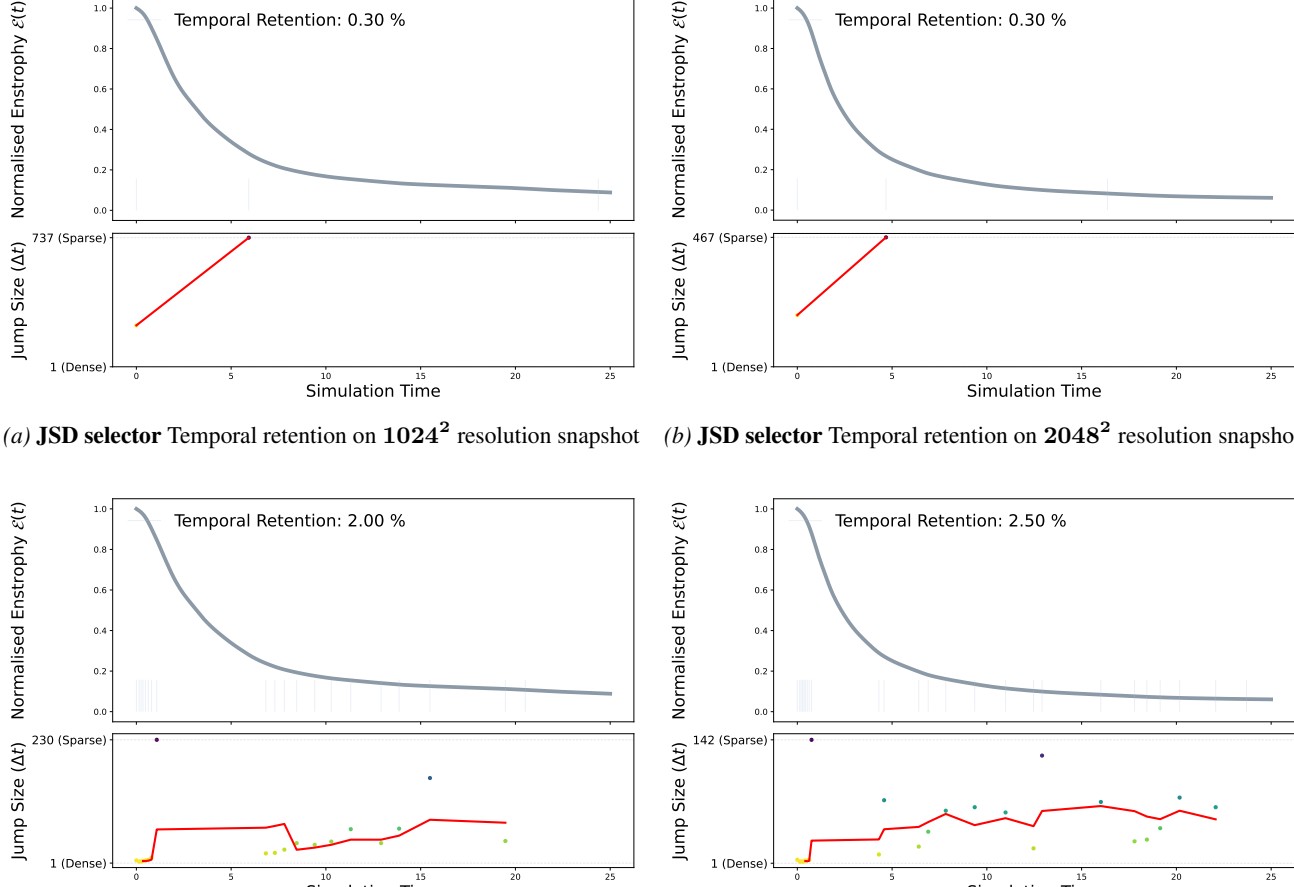

*(a)* **JSD selector** Temporal retention on $1024^2$ resolution snapshot   *(b)* **JSD selector** Temporal retention on $2048^2$ resolution snapshot

*(a)* **Spectral selector** Temporal retention on $1024^2$ resolution snapshot   *(b)* **Spectral selector** Temporal retention on $2048^2$ resolution snapshot

From a performance standpoint, the Residual entropy selector performs similar to our PATS and also retains the salient snapshots, i.e. denser selection in the high turbulence regime and modulates to coarser selection in the equilibrated regime and its temporal retention falls in a similar range of $31\% - 40\%$ for resolutions $\{256, 512, 1024\}$ as reported in Table 8.

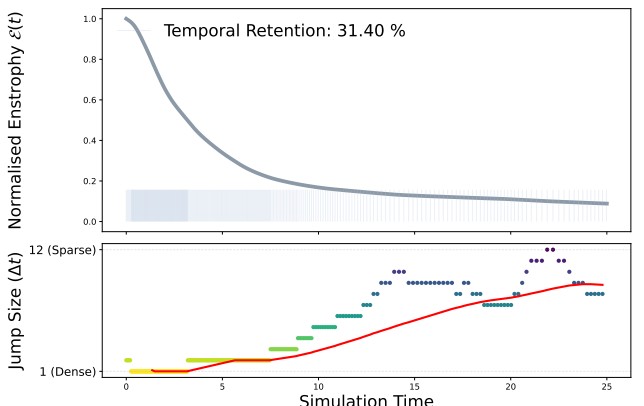

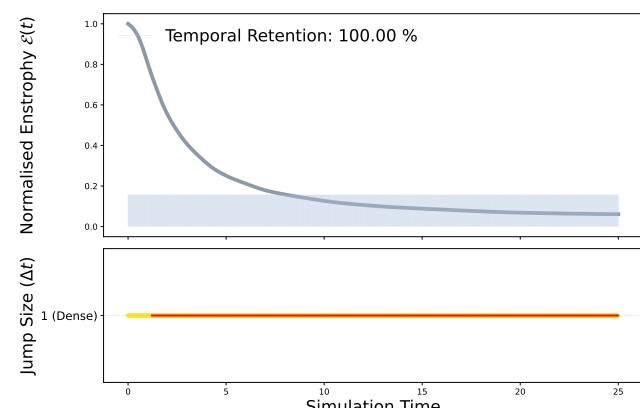

*(a)* **Residual selector** Temporal retention on $1024^2$ resolution snapshot

*(b)* **Residual selector** Temporal retention on $2048^2$ resolution snapshot

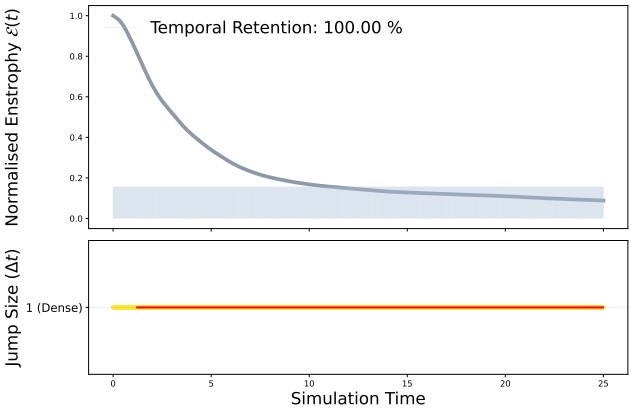

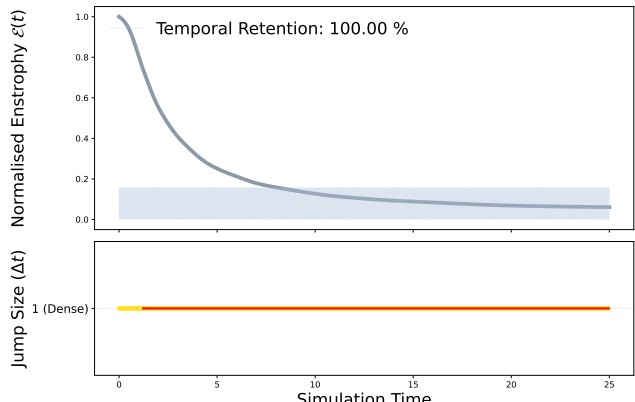

*(a)* **MI selector** Temporal retention on $1024^2$ resolution snapshot

*(b)* **MI selector** Temporal retention on $2048^2$ resolution snapshot

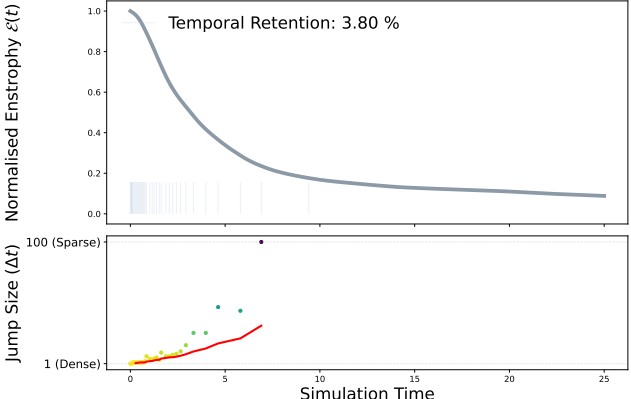

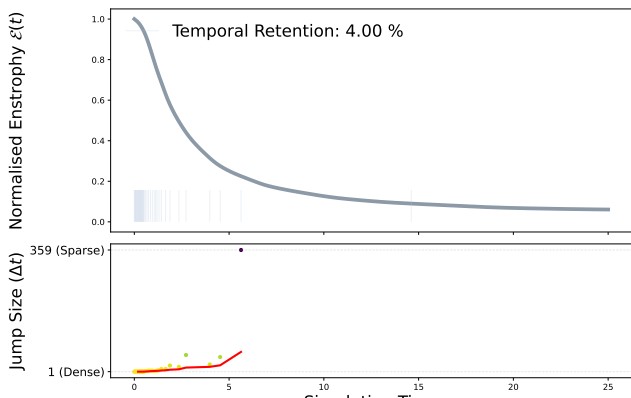

*(a)* **Momentum-aware selector** Temporal retention on $1024^2$ resolution snapshot

*(b)* **Momentum-aware selector** Temporal retention on $2048^2$ resolution snapshot

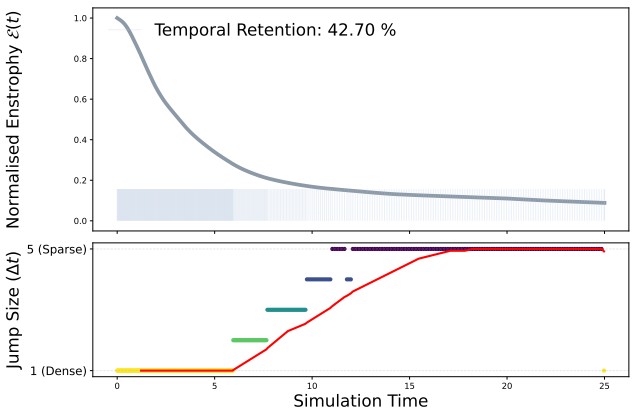 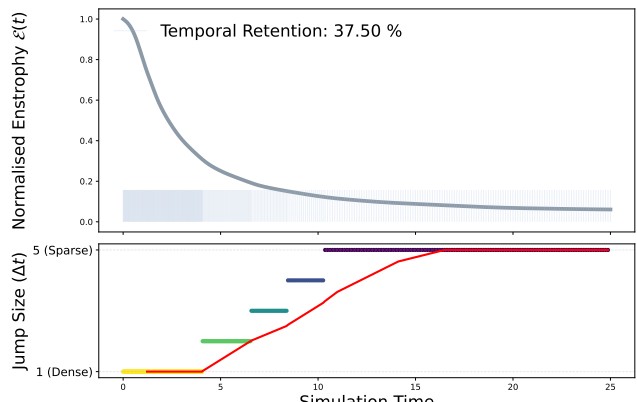

*(a)* **Enstrophy selector** Temporal retention on $1024^2$ resolution snapshot

*(b)* **Enstrophy selector** Temporal retention on $2048^2$ resolution snapshot

*Table 8.* **Temporal selector retention analysis across resolutions for 2D Kolmogorov flows.** Comparison of snapshot retention rates across temporal selectors and spatial resolutions $\{256^2, 512^2, 1024^2, 2048^2\}$ for a trajectory of 1000 snapshots. Retention is computed as $(selected frames/1000) \times 100\%$. The enstrophy-based PATS selector retains 37.5%–42.9% of snapshots consistently across all resolutions, reflecting a physically meaningful and resolution-stable selection that concentrates retained frames in high-turbulence transient regimes while discarding redundant snapshots during quasi-static evolution. In contrast, the physics-agnostic selectors exhibit two distinct failure modes: *under-retention*, where JSD (0.30% across all resolutions), Spectral (2.0%–2.7%), and Momentum (2.8%–5.5%) selectors discard the vast majority of snapshots indiscriminately, failing to identify dynamically critical transients; and *over-retention*, where MI retains all 1000 snapshots at every resolution (100%), providing no compression of the temporal dimension whatsoever. The Residual selector exhibits resolution-dependent instability, retaining 31.4%–39.4% at coarser resolutions but collapsing to full retention (100%) at $2048^2$, indicating sensitivity to the spectral content of high-resolution fields that renders it unreliable for in-situ deployment. Collectively, these results demonstrate that physics-agnostic selectors fail to generalize across resolutions and dynamical regimes, whereas PATS provides stable, physically grounded temporal compression that adapts to the underlying simulation dynamics.

| TEMPORAL SELECTOR | RESOLUTION | RETAINED FRAMES | RETENTION (%) |
|---|---|---|---|
| **ENSTROPHY** (PATS) | 256 | 429 | 42.90 |
| | 512 | 427 | 42.70 |
| | 1024 | 427 | 42.70 |
| | 2048 | 375 | 37.50 |
| **JSD** | 256 | 3 | 0.30 |
| | 512 | 3 | 0.30 |
| | 1024 | 3 | 0.30 |
| | 2048 | 3 | 0.30 |
| **RESIDUAL** | 256 | 394 | 39.40 |
| | 512 | 388 | 38.80 |
| | 1024 | 314 | 31.40 |
| | 2048 | 1000 | 100.00 |
| **SPECTRAL** | 256 | 27 | 2.70 |
| | 512 | 27 | 2.70 |
| | 1024 | 20 | 2.00 |
| | 2048 | 25 | 2.50 |
| **MI** | 256 | 1000 | 100.00 |
| | 512 | 1000 | 100.00 |
| | 1024 | 1000 | 100.00 |
| | 2048 | 1000 | 100.00 |
| **MOMENTUM** | 256 | 28 | 2.80 |
| | 512 | 55 | 5.50 |
| | 1024 | 38 | 3.80 |
| | 2048 | 40 | 4.00 |

Although, it breaksdown due to the following reasons: (i) It is heavily-resolution dependent, i.e. pixel-level artifacts can hughly affect these selectors, as a consequence yields $100\%$ Fig. 14b snapshot selection, thus sending redundant frames into the spatial neural compression module. This needs to be modulated using a tolerance, which is not known apriori in our in-situ scenarios, and (ii) These have jump size of 12 (see Fig. 14a), leading to loss in temporal coherence, required for global reconstruction, especially post-hoc analysis and relevant downstream tasks.

## I. Solver vs Neural Compression Time Scaling Laws

Traditional PDE solvers scale super-linearly with spatial resolution: advancing a single timestep requires applying numerical stencils, enforcing boundary conditions, and satisfying CFL stability constraints across all $N^d$ mesh points, resulting in wall-clock costs that grow as $\mathcal{O}(N^d)$ or worse for multi-scale or adaptive mesh refinement (AMR) schemes. Neural field training, by contrast, operates on a fixed-capacity implicit network whose parameter count is decoupled from the resolution of the underlying grid — increasing resolution enlarges the coordinate-value training set but does not alter the network architecture, yielding a substantially sub-linear growth in training overhead. As evidenced in Table 9 and Fig. 18, solver latency for 2D kolmogorov flows increases by approximately $380\times$ from $256^2$ to $2800^2$, whereas neural compression (NC) training time grows by only $\sim 3\times$ over the same range, reducing the NC-to-solver ratio from $7323\times$ at $256^2$ to $58\times$ at $2800^2$. This convergent scaling behavior implies that the relative overhead of neural compression diminishes systematically as resolution increases, making in-situ neural compression increasingly competitive for high-dimensional, mesh-intensive HPC simulations and multi-scale physics PDEs where solver costs dominate the total simulation budget.

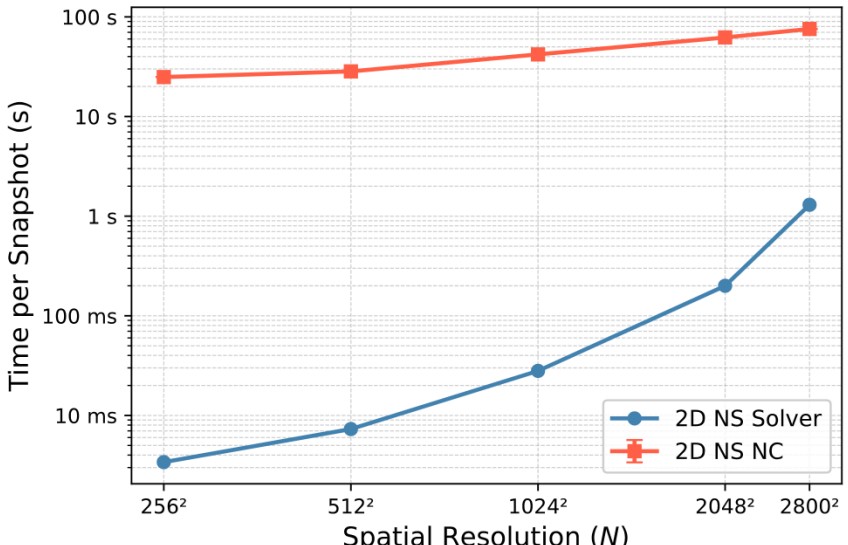

*Figure 18.* **Solver latency vs. neural compression scaling for 2D Kolmogorov flows.** Wall-clock time per snapshot plotted against spatial resolution $N^2$ for the traditional Navier-Stokes solver and `ANTIC` neural compression (NC) on a single NVIDIA H200 GPU. Solver latency scales super-linearly with resolution, consistent with $\mathcal{O}(N^2)$ mesh-point complexity under CFL constraints, while NC training time grows sub-linearly due to the resolution-invariant capacity of the underlying implicit neural field. The narrowing NC-to-solver ratio.

*Table 9.* **Hardware Benchmarks: Solver Latency vs. Neural Compression (NC).** Wall-clock training time per snapshot ($dt$) for 2D Navier-Stokes Kolmogorov flows. All benchmarks were performed on a single **NVIDIA H200** CUDA GPU. The scaling ratio (NC/Solver) demonstrates that while NC has a higher absolute latency, its complexity scales sub-linearly relative to the exponential growth of the traditional solver.

| RESOLUTION ($N^2$) | TOTAL PIXELS | SOLVER ($dt$) | NC (PER SNAPSHOT) | NC/SOLVER RATIO |
|---|---|---|---|---|
| $256^2$ | 65,536 | 3.4 MS | 24.90 s $\pm$ 1 MS | $\sim 7323\times$ |
| $512^2$ | 262,144 | 7.3 MS | 28.37 s $\pm$ 4 MS | $\sim 3886\times$ |
| $1024^2$ | 1,048,576 | 28 MS | 42.02 s $\pm$ 16 MS | $\sim 1500\times$ |
| $2048^2$ | 4,194,304 | 0.2 s | 62.19 s $\pm$ 22 MS | $\sim 310\times$ |
| $2800^2$ | 7,840,000 | 1.3 s | 75.67 s $\pm$ 41 MS | $\sim 58\times$ |

## J. Dynamic Adaptive Regulator vs Binary Adaptive Regulator

The PATS component utilizes a dynamic regulator to determine the optimal stride $\Delta\tau$ within the interval $[1, W]$. This approach enables fine-grained adaptive sampling by mapping the physics-derived saliency $\phi_W$ (physics metrics stored in the Queue with size $W$) to a discrete temporal jump $\tau_{n+1} - \tau_n \in \{1, \ldots, W\}$. For ablation and comparative analysis, we further implement a binary regulator, which is predominantly a "bang-bang" control variant that strictly switches between dense ($\Delta\tau = 1$) and sparse ($\Delta\tau = W$) sampling. This binary mode-switching is governed by a thresholding function of the localized physics metrics $f(\phi_W)$ over the window $W$:

$$t_{n+1} = \begin{cases} t_n + W & \text{if } f(\phi_W) \leq \gamma \\ t_n + 1 & \text{otherwise} \end{cases}$$

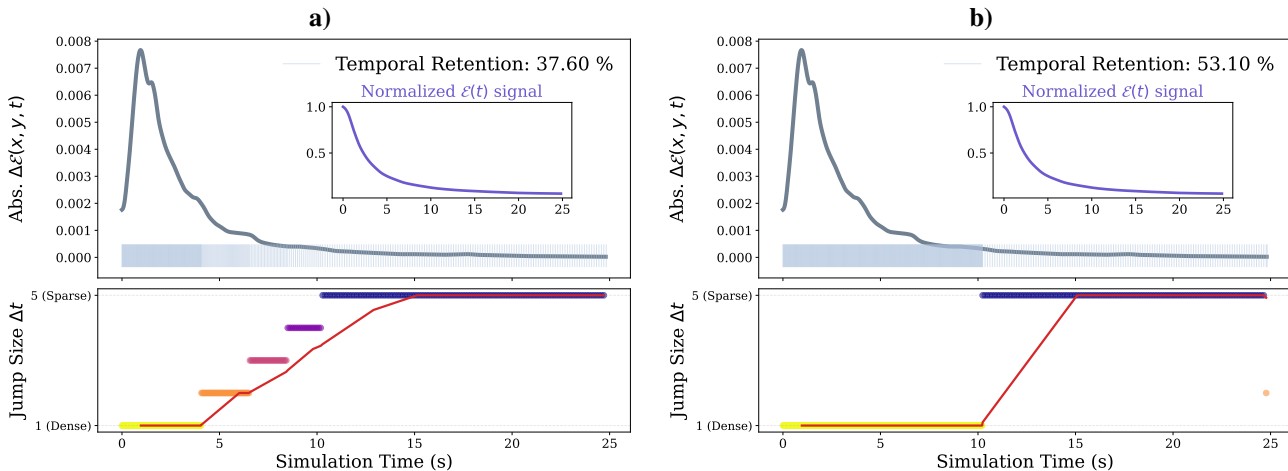

*Figure 19.* **Temporal retention comparison between dynamic vs binary regulator.** (Left) illustrates the comparative efficiency of our two regulation regimes. The Dynamic Regulator (Left) achieves a retention of $37\%$ by mapping physical saliency to a multi-step stride set $\{1, \ldots, 5\}$. Intermediate jumps (sizes 2, 3, and 4) allow the selector to smoothly transition between stationary and transient regimes, isolating the high-enstrophy activity with minimal redundancy. (Right) Conversely, the Binary Regulator utilizes a "Bang-Bang" strategy, restricted to either dense ($\Delta\tau = 1$) or maximum-stride ($\Delta\tau = 5$) sampling. While it successfully captures the high-enstrophy transients, the lack of intermediate quantization leads to over-sampling in moderate-activity regions, resulting in a higher temporal retention of $53\%$.

## K. Hardware and Licenses

**Computing Infrastructure**. Our primary computational experiments were conducted on two high-performance configurations. CPU-intensive tasks, including initial solver execution and physics-informed scalar extraction, were performed on either a dual-socket Intel Xeon Platinum 8452Y+ (64 total cores) clocked at 4.1 GHz with 2048 GiB of RAM, or a dual-socket Intel Xeon 6767P (128 total cores) at 2.4 GHz equipped with 3072 GiB of RAM.

Simulation solvers, neural field training and high-fidelity snapshot processing were accelerated using either an NVIDIA H200 SXM GPU (144 GiB HBM3e) or a next-generation NVIDIA B300 SXM6 GPU (275 GiB). All GPU-accelerated workloads utilized CUDA 13.1 and NVIDIA Driver `590.48.01`, ensuring optimal support for the Blackwell architecture's enhanced precision and throughput.

**Licenses and Software**. This work would not have been possible without the open-source software ecosystem. Our implementation is built upon multiple community-maintained libraries, and we gratefully acknowledge their licenses below. The core computations were performed using `JAX[cuda12]` (Bradbury et al., 2018) with `CUDA` support, licensed under the Apache 2.0 License. For model definition and training, we relied on `Flax NNX`, `Orbax` and `Optax`, both also under Apache 2.0. All libraries used are permissively licensed, enabling free academic and non-commercial research.

