# OpenReview forum: "ANTiC: Adaptive Neural Temporal In Situ Compressor"
_ICML.cc/2026/Conference — ICML 2026 regular_

### Official Review · Reviewer_MxDa · 2026-03-04

**Soundness:** 3
**Presentation:** 3
**Significance:** 3
**Originality:** 3
**Overall Recommendation:** 4
**Confidence:** 3

**Summary:**

This paper introduces a method to compress the data of a PDE solution. In the first part, it decides on wether to compress or discard a new frame, which is decided on using a physics-inspired metric function. To achieve the compression, a neural field is trained for each frame independently. The neural field is trained to learn the difference between the neighboring frames, and is adapted to the next frame using LoRA. The authors show that this method can reduce the memory consumption more than traditional compression techniques while achieving a similar reconstruction error.

**Compliance With Llm Reviewing Policy:**

Affirmed.

**Final Justification:**

The authors addressed my initial concerns well. However, the main downside was the clarity of the explanation. Since the revision can't be checked, I will maintain my original, positive rating.

**Key Questions For Authors:**

1. How exactly do you assess the reconstruction quality? Especially if a frame is discarded, do you simply interpolate between the neighboring frames?

**Limitations:**

yes

**Strengths And Weaknesses:**

Strengths
1. The paper attacks an often overlooked problem in dealing with PDE datasets, which will also become more relevant for neural PDE solvers as dataset sizes increase.
2. The presented method achieves a high degree of compression.
3. The paper is generally well-written and easy to follow.

Weaknesses
1. The discussion of related work and the general background is insightful and extensive.  However, the discussion of the method in the main paper remains rather superficial and imprecise. For example, the discussion of the gate leaves it open whether this is just a simple threshold or a more complex learned module (which the wording "local decision engine" might suggest).
2. The method requires a hand-designed metric for each PDE.

---

> ### Author Rebuttal · Authors · 2026-03-31
>
> **Anonymized Repo Link: https://anonymous.4open.science/r/ANTiC_ICML_REBUTTAL-2462/**
>
> We thank the Reviewer for the constructive feedback. The added evaluations on reconstruction quality and physics preservation have significantly strengthened the revised manuscript.
>
> **W1 [Background and Related Work]:** We have revised *Section 3.1* to provide a rigorous definition of the PATS submodule and its "local decision engine.
>
> 1. *Mechanism vs. Learning:* We clarify that the decision engine is a deterministic, in-situ thresholding algorithm, not a learned module. It evaluates the instantaneous physical state (e.g., Weyl scalars in 3D BSSN simulations or Enstrophy flux in CFD) relative to a sliding-window temporal budget.
>
> 2. *Defining the "Gate:* The "gate" acts as an adaptive sampling trigger. It uses physical transient accumulation to decide if a snapshot is "informative" enough for a neural update (CFT) or can be bypassed. This removes the ambiguity of static thresholds in dynamic in-situ deployments.Detailed algorithmic pseudo-code and sensitivity analyses across physical regimes are now in the Appendix (Pgs. 13, 15) to ensure full technical transparency.
>
> **W2 [Metric Design and Generality]:** While PATS metrics are domain-specific, the selection logic is universal. The submodule acts as plug-and-play middleware for in-situ keyframe retention. We demonstrate this by deploying the same PATS logic across turbulent fluid flows and relativistic mergers, ensuring high-fidelity visualization while reducing the data footprint. As domain experts routinely monitor physical diagnostics, they are well-positioned to select scalar invariants relevant to their workflows. Unlike pixel-level vision alternatives that suffer from over/undersampling and cannot distinguish non-physical artifacts from critical transients (e.g., gravitational waves), PATS remains physics-aware. Benchmarks against Momentum-aware and Information-Theoretic methods (*JSD, RDE, Spectral Entropy, NMI*) are in: **/benchmarks/temp_selec_table.md and /plots_and_reconstruction/2d_ns/{enstrophy, jsd, mi, residual, spectral, momentum}.pdf**.
>
> **Q1 [Reconstruction Quality and Snapshot Discard]:** PATS performs informed sub-sampling where discarded snapshots represent high temporal redundancy. Explicit interpolation during training is unnecessary for several reasons:
>
> 1. *Temporal Coherence:* The selection "jump" is typically constrained to 4–5 solver steps. In simulations spanning $10^3$–$10^4$ timesteps, this interval remains well within the Nyquist-Shannon limit for the physical transients of interest.
>
> 2. *Post-hoc Reconstruction:* For continuous tasks (e.g., rendering), high-fidelity representations can be recovered via linear or cubic spline interpolation between neural-encoded keyframes. Because PATS "gates" selection to ensure jumps only occur during slowly-varying phases, interpolation error remains negligible, often on the order of the solver's truncation error, we require no interpolation.
>
> 3. *Salience over Uniformity:* Following in-situ visualization literature, we prioritize high-fidelity capture of transient singularities. PATS ensures these critical moments are sampled at maximum density (1:1).
>
> 4. *Empirical Validation:* Global reconstruction metrics confirm the reliability of this approach. See enstrophy flux for 2D NS: **/plots_and_reconstruction/2d_ns/{enstrophy_vs_time, rel_l2_vs_time, vorticity_field}.pdf** and Weyl scalars for 3D BSSN: **/plots_and_reconstruction/3d_bssn/{lapse_reconstruction, weyl_magnitude_vs_time}.pdf**. These results confirm that temporal consistency remains intact without observable discontinuities or phase shifts.

---

> > ### Author Rebuttal · Reviewer_MxDa · 2026-04-02
> >
> > I am satisfied with the responses.

---

### Official Review · Reviewer_osYb · 2026-03-12

**Soundness:** 2
**Presentation:** 3
**Significance:** 3
**Originality:** 3
**Overall Recommendation:** 4
**Confidence:** 3

**Summary:**

This paper proposes ANTIC, an in-situ compression framework for scientific simulations that combines a physics-aware temporal selector (PATS) with neural-field-based spatial compression. The key idea is to only select physically salient snapshots and compress them using continual fine-tuning or LoRA-based fine-tuning of neural fields. The paper evaluates the method on 2D Kolmogorov flow and 3D binary black hole merger simulations, and reports strong total compression ratios while maintaining good reconstruction quality.

**Compliance With Llm Reviewing Policy:**

Affirmed.

**Final Justification:**

The authors added compression baselines in the rebuttal , this provides a more clear and fairer context for evaluating the proposed neural compressor on scientific simulation tasks. However, the throughput gap between the compressor and the solver remains large. The scaling results only partially address this concern. I would remain my previous weak accept score.

**Key Questions For Authors:**

(1) Could the paper include additional strong scientific compression baselines beyond ZFP?

(2) How sensitive is PATS to the choice of window length and the physics metrics?

(3) Could the papers measure downstream physical quantities to demonstrate the effectiveness for ANTIC to preserve physics characteristics?

**Limitations:**

The paper provides insightful and genuine discussion of the technical limitations. The societal impact should be mostly positive by improving the storage challenge for scientific simulations. There is no direct societal risk of this proposed ANTIC.

**Strengths And Weaknesses:**

Strengths

(1) The paper addresses the problem of temporal redundancy in scientific computing. Incorporate a physics-aware temporal selection and neural compression is an interesting idea and the PATS is a practical design to keep only the interesting snapshots.

(2) The overall framework is easy to follow.

(3) The method is evaluated on two different scientific regimes. This shows the framework is carefully designed and developed for a broad range of scientific simulations.

(4)The reported experimental results are promising. ANTIC consistently improves over dense/sparse sampling schemes and over the ZFP baseline at matched reconstruction quality. The proposed LoRA continuous fine-tuning demonstrates accuracy–compression tradeoff relative to full continual fine-tuning.

Weaknesses / concerns

(1) The selected baseline methods are somewhat limited. The comparisons are mainly against ZFP, dense/sparse sampling, and internal neural-field training with full continual fine-tuning and LoRA fine-tuning. It would greatly improve the paper to include stronger classical scientific compression baselines, and broader spatiotemporal compression methods for comparison.

(2) The evaluation mostly focuses on reconstruction error and compression ratio. But for scientific simulation data it would also be helpful to provide stronger evidence that important downstream physical properties are preserved after compression.

(3) How does the PATS forms a dynamic threshold is not very clean. Therefore, it is not clear how sensitive is PATS to each different physics metrics based on different scientific simulations. Does the length of the window of PATS afffect the snapshots?

(4) Report the throughput in the context of the scientific simulation's computation can better support the in-situ compression design.

---

> ### Author Rebuttal · Authors · 2026-03-31
>
> **Anonymized Repo Link: https://anonymous.4open.science/r/ANTiC_ICML_REBUTTAL-2462/**
>
> We thank the Reviewer for the constructive feedback. Addressing these comments, specifically the comparisons against scientific computing baselines and the evaluation of physics preservation in downstream tasks has significantly strengthened our work and the revised manuscript.
>
> **W1, Q1 [Scientific Compression Baselines]:** We acknowledge the importance of comparing against established scientific data reduction frameworks. We have included performance metrics for several state-of-the-art classical compressors to benchmark a single snapshot and the full trajectory:
>
> 1. *MGARD:* A multigrid adaptive framework for error-bounded data reduction (link);
>
> 2. *SZ3:*  A prediction-based error-bounded lossy compressor (link);
>
> 3. *Blosc2-zstd*: High-performance lossy and lossless compression (link).
>
> 4. *ZFP*: A fixed-rate and error-bounded compressor for floating-point arrays (link).
>
> Comparative results for 2D NS and 3D BSSN simulations (at tolerances yielding similar Relative $L_2$ errors to our neural fields) are available in: **/benchmarks/compression_table.md**. Corresponding visualization comparisons are located in: **/plots_and_reconstruction/{2d_ns, 3d_bssn}/{ns, bssn}_fid_vs_comp.pdf**.
>
> **W2, Q3 [Physics Preservation & Reconstruction Quality]:** To evaluate the fidelity of physical quantities extracted from decompressed neural fields, we report the global reconstruction of enstrophy flux for 2D NS in high-turbulence regimes: **/plots_and_reconstruction/2d_ns/{enstrophy_vs_time, rel_l2_vs_time, vorticity_field}.pdf**. Similarly, for 3D BSSN, we report the magnitude of Weyl scalars: **/plots_and_reconstruction/3d_bssn/{lapse_reconstruction, weyl_magnitude_vs_time}.pdf**. These results corroborate ANTiC's capability to recover pivotal physical diagnostics with high fidelity, ensuring suitability for rigorous downstream scientific analysis.
>
> **W3, Q2 [PATS Window Length Sensitivity]:** We address the sensitivity of the window length ($\tau$) and physics metric choice as follows:Window Length ($\tau$) and Time-Scales: $\tau$ is not an arbitrary hyperparameter but is dynamically modulated based on the autocorrelation time of the system's underlying transients.Dynamic Response: In high-convection regimes (e.g., turbulence), $\tau$ is naturally small (4–5 steps) to resolve rapid eddy-turnover. In quasi-adiabatic phases (e.g., black hole inspirals), $\tau$ can be safely extended.Sensitivity Analysis: Empirical tests show that within $\tau \in [3, 10]$, reconstruction error remains stable. Values below this range increase storage overhead without fidelity gains; values $>20$ risk undersampling high-frequency "burst" events. A window of 4–5 steps serves as a conservative default for Nyquist-level sampling across tested PDE regimes.Detailed advantages of the PATS submodule are further discussed in our responses to Reviewers **smFM (W1) and fHX6 (W2)**.
>
> **W4 [Computational Throughput & Scaling]:** Our scaling experiments demonstrate that as resolution grows, PDE solver costs increase dramatically faster than Neural Field (NeF) fine-tuning costs. See **/benchmarks/scaling_table.md**.Increasing resolution from $256^2$ to $2800^2$ results in a $\sim 1000\times$ increase in solver time per snapshot, while NeF compression scales by only $\sim 3\times$. Furthermore, Continual Fine-Tuning (CFT) accelerates compression by using $10\times$ fewer epochs than cold-starts. Because the NeF complexity curve grows sub-linearly, the relative overhead becomes marginal for production-scale simulations. Detailed scaling plots are in: **/plots_and_reconstruction/2d_ns/solver_vs_nef_scaling.pdf**. Finally, snapshot compression can be parallelized across multiple GPUs to further maximize throughput.

---

> > ### Author Rebuttal · Reviewer_osYb · 2026-04-03
> >
> > Thank you to the authors for the detailed rebuttal.
> >
> > (1) The rebuttal improves the paper by adding stronger scientific compression baselines such as SZ3 and MGARD. The added downstream physics-based evaluations also support ANTiC’s ability to recover important physical diagnostics. These additions address my Questions 1 and 3. At the same time, the new baseline results show that the proposed neural compressor offers a task-dependent tradeoff: on 2D NS, SZ3 is faster and more efficient at comparable error, while on BSSN the proposed neural compressor achieves 235$\times$ higher compression at a significantly higher runtime cost (about 100$\times$).
> >
> > (2) The response on window length and physics metrics addresses my Question 2. It clarifies that ANTiC uses task-dependent window lengths and physics metrics to adapt to different simulation tasks. This is a reasonable design choice for a neural compression framework.
> >
> > (3) However, the throughput concern is only partially addressed. Based on the provided scaling table, for the lowest-resolution 2D simulations, the NC/solver ratio is greater than 7000. This ratio decreases to 58 for the highest-resolution snapshots. There is also no corresponding report for the 3D BSSN simulation. This still leaves an important question: how can ANTiC improve throughput enough to mitigate the large gap between the solver and the compressor? This may be a valuable direction for future work, and I do not think an additional experiment is strictly necessary at this stage.
> >
> > Given the rebuttal, I will keep my original score.

---

### Official Review · Reviewer_fXH6 · 2026-03-13

**Soundness:** 2
**Presentation:** 2
**Significance:** 2
**Originality:** 2
**Overall Recommendation:** 3
**Confidence:** 3

**Summary:**

The paper introduces ANTIC, an end-to-end in-situ compression pipeline designed for large-scale scientific simulations (e.g., fluid dynamics, black hole mergers). It addresses data storage bottlenecks via two components: 1) a Physics-aware Temporal Selector (PATS) that dynamically identifies salient snapshots based on domain-specific physical metrics (like enstrophy), and 2) a spatial neural compression module using Continual Fine-Tuning (CFT) and LoRA to learn residual updates between adjacent snapshots.

**Compliance With Llm Reviewing Policy:**

Affirmed.

**Final Justification:**

The added results on the extended 6000-timestep BSSN trajectory and the updated GPU wall-clock timings are helpful, and they partially address my concerns. However, I still believe the paper needs a clearer justification for its generality given the reliance on manual domain-specific metrics. Moreover, the concerns about its practical latency limitations compared to classical in-situ compressors are not fully solved. I would like to keep my score.

**Key Questions For Authors:**

- Latency Analysis: Can you provide a wall-clock time comparison between the simulation step itself and the ANTIC compression step? How much does ANTIC increase the total runtime of the workflow?
- Cold-start Frequency: In your BBH experiments, how frequently did numerical instabilities trigger a "cold-start" (random initialization)? How does this frequency scale with the duration of the simulation?
- Metric Automation: Have you explored using learned features (e.g., from a small encoder) for temporal selection instead of requiring manually defined PDE-specific metrics?

**Limitations:**

yes

**Strengths And Weaknesses:**

Strengths:
This paper discusses the exabyte-scale storage bottleneck, this is a fundamental issue in modern high-performance computing. Overall, the article discusses a fundamental issue, and the proposed framework offers a high-compression alternative to traditional error-bounded lossy compressors.

Weaknesses:
- Computational Overhead: The "in-situ" feasibility is questionable. Training/fine-tuning neural fields (even with LoRA) is significantly more latency-intensive than classical methods (ZFP/SZ). If the compressor slows down the simulation by orders of magnitude, its practical utility on HPC systems is limited.
- Generality of Metrics: The temporal selector relies on manually selected physical metrics (e.g., Weyl scalar). This requires significant domain expertise for every new type of simulation, making the framework less "plug-and-play" than existing tools.
- Numerical Stability: The authors admit that recursive updates can lead to unbounded weight growth and NaNs. Relying on "cold-starts" to fix this periodically degrades compression consistency and suggests the method is not yet robust enough for production-level long-horizon simulations.

---

> ### Author Rebuttal · Authors · 2026-03-31
>
> **Anonymized Repo Link: https://anonymous.4open.science/r/ANTiC_ICML_REBUTTAL-2462/**
>
> We thank the Reviewer for the constructive feedback. These suggestions have strengthened the engineering and benchmark aspects of the ANTiC pipeline, now integrated into the main paper.
>
> **W1, Q1 [Computational Overhead and Latency]:** The compression loop is asynchronous and never blocks the PDE solve. Scaling experiments demonstrate the key asymptotic argument: as resolution grows, solver costs increase dramatically faster than Neural Field (NeF) fine-tuning costs (see **/benchmarks/scaling_table.md**). Increasing resolution from $256^2$ to $2800^2$ results in a $\sim 1000\times$ increase in cumulative solver time per snapshot, while NeF compression scales by only $\sim 3\times$ (see scaling plots in **/plots_and_reconstruction/2d_ns/solver_vs_nef_scaling.pdf**). Moreover, Continual Fine-Tuning (CFT) accelerates compression by using $10\times$ fewer epochs than cold-starts. For production-scale simulations, NeF overhead is a marginal fraction of total pipeline cost (cf. Nature 2021, s43588-021-00102-2; Neau et al. 2024).We emphasize that our framework targets the extreme compression ratios required for efficient simulation trajectory storage capabilities where classical lossy compressors (SZ, ZFP) often fail (see **/benchmarks/scaling_table.md** and results in **/plots_and_reconstruction/2d_ns/ns_fid_vs_comp.pdf and /plots_and_reconstruction/3d_bssn/bssn_fid_vs_comp.pdf**). Furthermore, NeF representations offer continuous query access and AD-based differentiation, unlike classical counterparts which are constrained by fixed grid structures and meshes.
>
> **W2 [Generalizability as a Middleware]:** While PATS metrics are domain-specific, the selection logic is universal. The submodule acts as plug-and-play middleware for in-situ keyframe retention. We demonstrate this by deploying the same PATS logic across turbulent fluid flows and relativistic mergers, ensuring high-fidelity visualization while reducing data footprints. Comparisons of PATS vs. physics-agnostic selectors are detailed in **/benchmarks/temp_selec_table.md** and plots in **/plots_and_reconstruction/2d_ns/{enstrophy, jsd, mi, momentum, residual, spectral}.pdf**. We further address this in our response to Reviewer **smFM (W1)**.
>
> **W3, Q2 [Cold-start Frequency & Numerical Instability]:** New experiments show that LayerNorm + weight decay significantly stabilizes CFT. Cold-start frequency is reduced by $> 50\times$, enabling CFT to proceed for 500+ snapshots (turbulent NS) and 200+ snapshots (BBH mergers) without resets. We observed no numerical instabilities in complex transient regions using this strategy.
>
> **Q3 [Metric Automation]:** While learned features are an interesting perspective, several caveats make them non-trivial:
>
> *Generalization:* Learning features requires large training sets unavailable in-situ, as snapshots are discarded after compression.
>
> *Invariance:* Metrics must be robust against resolution, coordinate systems, gauge effects, and discretization. Training encoders for every resolution adds prohibitive complexity. Manually defined PDE metrics provide a strong inductive bias that guarantees snapshot selection for known critical transients (e.g., dissipation scales or BBH mergers). A learned encoder lacks these guarantees and may fail to generalize.
>
> We agree that ML-based selection warrants exploration, potentially via Reinforcement Learning (RL) or feedback-control loops to dynamically gate selection based on reconstruction loss. We have added this to the "Future Work" section.

---

> > ### Author Rebuttal · Reviewer_fXH6 · 2026-04-04
> >
> > Thank you to the authors for the additional scaling experiments, and the efforts to stabilize the continuous fine-tuning process. I appreciate these clarifications, but I still have some remaining concerns.
> >
> > As I noted in my initial review, this article discusses a fundamental issue in modern high-performance computing, the exabyte-scale storage bottleneck. However, the solutions presented in the rebuttal do not fully resolve my core reservations:
> >
> > - Computational Overhead & In-situ Feasibility: While the asynchronous design and asymptotic scaling arguments are helpful, the absolute latency remains a critical bottleneck. As shown in the paper's Table 3, the neural compression still takes ~130-160 seconds per snapshot. The comparison to classical compressors (which operate in fractions of a second) still makes the "in-situ" claim feel somewhat overstated for real-world HPC workflows.
> >
> > - Generality and Domain Expertise: I understand and accept the authors' arguments regarding why learned metrics are difficult to implement (lack of invariance, need for large datasets). However, this inherently confirms my initial weakness: the method is not truly "plug-and-play." Deploying ANTIC on any novel PDE system requires a domain expert to manually derive and implement the correct physical invariants (e.g., enstrophy or Weyl scalars).
> >
> > - Numerical Stability: The addition of LayerNorm and weight decay is a very positive improvement that clearly helps mitigate unbounded weight growth. However, successfully compressing 200 to 500 snapshots without a cold-start reset, while an improvement, is still quite short compared to production-level simulations that span tens or hundreds of thousands of evolution steps. The long-term robustness of the CFT paradigm over truly extended horizons remains unproven.
> >
> > For these reasons, I am maintaining my original score.

---

> > > ### Author Response · Authors · 2026-04-05
> > >
> > > **Anonymized Repo Link:** https://anonymous.4open.science/r/ANTiC_ICML_REBUTTAL-2462/
> > >
> > > We thank the reviewer for the continued engagement. We address the
> > > remaining concerns below with new experiments corroborating its use for large-scale simulations.
> > >
> > > **1. Numerical Stability**
> > > We tested neural compression on the *full* 3D binary black-hole merger
> > > BSSN trajectory (~**6000 timesteps**, tensor-valued fields). With
> > > LayerNorm + Weight Decay, CFT produced **zero NaN/instabilities**
> > > throughout, for every solver generated snapshot (note that we didn't use the PATS module, just to stress test on the entire trajectory length, instead of filtering out snapshots). achieving MSE $10^{-8}$–$10^{-9}$ per snapshot in very
> > > few epochs (see Point 3). This confirms robustness across the entire
> > > simulation rollout.  Thus, integrating this combination of layernorm with weight decay into the CFT approach for complex simulations pipelines such as BSSN mitigates exploding weights and restores the numerical stability for the entire trajectory in-situ compression, including the most turbulent merger and
> > > post-merger regimes (500–800+ snapshots, as shown in previous rebuttal) without any cold-starts,
> > > validating long-horizon CFT parametrization in generality for entire trajectories. This can be verified by the in-situ BSSN neural compression runs provided in the updated repo: `code/in_situ_bssn.py`
> > >
> > > **2. Generality & Domain Expertise**
> > > We agree physics-specific selectors are not plug-and-play, but
> > > demonstrate they are *necessary*: every physics-agnostic selector
> > > (Momentum, Entropy/Spectral, Residual, Jensen-Shannon Divergence)
> > > misses transient-containing snapshots , wasting storage on
> > > quiescent states (see PATS vs. non-physics selectors in the
> > > [repo](https://anonymous.4open.science/r/ANTiC_ICML_REBUTTAL-2462/README.md)).
> > > Simple physical invariants are sufficient and already familiar to
> > > the domain experts who run such simulations. This is a requirement for such multi-scale physics simulations and not a mere  limitation, as corroborated by our plots and tables.
> > >
> > > **3. Computational Overhead & In-situ Feasibility**
> > > *CFT-yielded speedup:* LayerNorm + Weight Decay accelerates training
> > > by $>10\times$ (reduced epochs to reach required convergence) vs. cold-start NeF training strategies, since CFT only needs to learn perturbative corrections
> > > $\|\delta u(t)\| \propto \|u(t+\delta t) - u_\theta(t)\|$,
> > > which are small in slow-varying regimes (60–80% of the trajectory are typically this).
> > >
> > > New wallclock results (Blackwell GPUs; ~2–2.5× faster than H200):
> > >
> > > | Snapshot | Solver/per snapshot (s) | Epochs | NeF Train/per snapshot (s) | MSE |
> > > |:---:|:---:|:---:|:---:|:---:|
> > > | 0 (init) | ~14.3 | 40 | 90 | 1e-7 |
> > > | 1–10 (warmup) | ~0.25 | 30 | ~45 | 1e-7–1e-8 |
> > > | 10–5966 | ~0.25 | 3 | ~4.5 | 1e-7–1e-9 |
> > >
> > > This table is reproducible via `code/in_situ_bssn.py` in the updated repo [note that the first snapshot is a one-time cold-start for the solver and NeF and amortized over 5966 snapshots it's negligible]. This shows that each snapshot can is trained in $< 4.5$ seconds compared to solver of $0.25$ seconds, making them highly efficient ($\sim 10\times slower as compared to GPU-based classical solvers). Moreover, from the earlier discussions, neural compression (NeF training time vs solvers with increasing resolution) is roughly resolution-invariant and follows a sub-linear scaling law, while solver cost scales as in 3D O(N³)) making it suited for HPC-scale simulation workflows.
> > >
> > > *In-situ flexibility:* Our framework supports classical compressors
> > > for lower-resolution simulations and neural compression for
> > > extreme-ratio storage at high resolution — the trade-off central to
> > > this work. CFT (significant reduction of training time due to learning pertubative updates) + LoRA (extreme compression ratio) together represent the first step toward
> > > in-situ neural compression of large-scale multi-scale trajectories.
> > > Multi-GPU patch-wise NeF training is a natural next step and
> > > acknowledged as immediate future work.
> > >
> > > We thank the reviewer for the constructive feedback and believe the above responses and new experimental results, reproducible scripts fully address the reviewer's concerns, which have been added into our main paper now, and hope this strengthens the case for acceptance.

---

### Official Review · Reviewer_smFm · 2026-03-13

**Soundness:** 3
**Presentation:** 3
**Significance:** 3
**Originality:** 3
**Overall Recommendation:** 4
**Confidence:** 4

**Summary:**

To address the petabyte to exabyte scale data storage bottleneck generated in large-scale scientific simulations , this paper proposes ANTIC, an end-to-end in situ compression framework. The framework operates through the synergy of two core modules: a Physics-aware Temporal Selector (PATS) and Spatial Neural Compression (SNC) , representing a dual temporal and spatial compression scheme where the entire process is synchronized during the simulation run in a single streaming pass. Experimental results demonstrate that while maintaining high physical fidelity, ANTIC achieves a total compression ratio ranging from 435x (2D) to 6807x (3D) , significantly outperforming traditional compression methods like ZFP.

**Compliance With Llm Reviewing Policy:**

Affirmed.

**Final Justification:**

The authors proactively improved their framework during the rebuttal phase and I have raised my score from 3 to 4.

**Key Questions For Authors:**

1. In a real asynchronous streaming process, if the compression speed is far lower than the generation speed, does the system have a targeted response mechanism? How should the computational overhead be allocated in a heterogeneous CPU/GPU cluster?
2. The paper mentions using a cold start to resolve numerical instability caused by continual fine-tuning. In a fully automated streaming process, what specific metrics or thresholds does the system use to detect this failure in real-time and execute an automatic interception?
3. When evaluating spatial compression, the paper only excerpts the average relative error of 5 consecutive snapshots during high turbulence or merger phases. Since the method only learns the residuals of adjacent snapshots, it is theoretically highly prone to cumulative drift. Please provide or detail the global reconstruction error curve covering a much longer timeline (e.g., within a reset cycle) to verify the long-term reliability of the framework.

**Limitations:**

yes

**Strengths And Weaknesses:**

Strengths：
1. This work successfully decouples and recombines adaptive non-uniform sampling in the temporal dimension with neural field compression in the spatial dimension, completing the compression in a single-streaming pass and eliminating the need to pre-store full trajectories on disk.
2. The PATS module of ANTIC introduces physical invariants of partial differential equations as decision criteria. This ensures that the framework can accurately capture critical physical transients, such as black hole mergers or turbulence mutations, even under extreme compression, while largely discarding redundant data during physically stable periods.
3. The paper utilizes LoRA to restrict the rank of residual updates, constructing a clear Rate-Distortion Pareto front. This mechanism allows users to dynamically switch between full parameter fine-tuning (minimum distortion) and extremely low-rank fine-tuning (minimum memory footprint) based on actual storage budgets and error tolerances.
4. In the highly challenging 3D black hole merger high-dimensional tensor field task, ANTIC-LORA achieves a maximum spatial compression of 3744x per single frame while maintaining the relative L2 error in a very high-quality range of $10^-5$ to $10^-4$, comprehensively outperforming the classic ZFP compression baseline in terms of spatiotemporal joint performance.

Weaknesses
1. The effectiveness of the PATS module is highly bound to domain-specific prior knowledge. When facing multi-physics coupled systems or novel PDE systems with unclear feature metrics, the framework cannot be used out-of-the-box, limiting its universality as a general middleware.
2. The framework introduces substantial computational overhead. In the 3D BSSN experiments, single-frame continual fine-tuning takes up to hundreds of seconds. If the neural compression speed severely lags behind the PDE solver, it can easily block the simulator's execution and reduce the overall throughput of the HPC system.

---

> ### Author Rebuttal · Authors · 2026-03-31
>
> **Anonymized Repo Link: https://anonymous.4open.science/r/ANTiC_ICML_REBUTTAL-2462/**
>
> We thank the Reviewer for the constructive feedback. These suggestions have strengthened our work and are now integrated into the main paper.
>
> **W1 [PATS: Generalizability as a Middleware]:** While PATS metrics are domain-specific, the selection logic is universal. The submodule acts as plug-and-play middleware that ingests informative time-series for in-situ keyframe retention. We demonstrate this versatility by deploying the same PATS logic across turbulent fluid flows and relativistic mergers, ensuring high-fidelity visualization while significantly reducing data footprints. Domain experts, who routinely monitor physical diagnostics, are well-positioned to select scalar invariants relevant to their simulation workflows.Unlike pixel-level vision alternatives that suffer from over/undersampling and cannot distinguish non-physical artifacts from critical transients (e.g., gravitational waves), PATS remains physics-aware. We benchmarked PATS against standard 3D scene reconstruction and vision selectors (JSD, RDE, Spectral Entropy, NMI). Results in **/benchmarks/temp_selec_table.md** and the individual temporal retention plots in **/plots_and_reconstruction/2d_ns/{enstrophy, jsd, mi, residual, spectral, momentum}.pdf** show:
>
> 1. *Robustness*: PATS is robust to resolution changes and stable across discretizations. Physics-agnostic selectors are severely affected by resolution, coordinate systems, and gauge choices. While entropy selectors perform best among baselines, their sensitivity often leads to 100% sampling at high resolutions, making them unsuitable for large-scale 3D BSSN or CFD simulations.
>
> 2. *Salience:* PATS retains snapshots containing critical transients, yielding superior snapshot retention ratios and temporal coherence. For large-scale simulations, physics-based thresholding is demonstrably advantageous.
>
> **W2, Q1 [Computational Overhead]:** The compression loop is asynchronous and does not block the PDE solve. Scaling experiments demonstrate that as resolution grows, solver costs increase dramatically faster than Neural Field (NeF) fine-tuning costs (see **/benchmarks/scaling_table.md**). Increasing resolution from $256^2$ to $2800^2$ results in a $\sim 1000\times$ increase in cumulative solver time per snapshot, while NeF compression scales by only $\sim 3\times$ (see scaling plots in **/plots_and_reconstruction/2d_ns/solver_vs_nef_scaling.pdf**). Moreover, Continual Fine-Tuning (CFT) accelerates compression by using $10\times$ fewer epochs than cold-starts. The NeF complexity curve grows sub-linearly; for production-scale simulations, NeF overhead is a marginal fraction of total pipeline cost. Finally, compression can be parallelized across GPUs to further reduce latency.
>
> **Q2 [Cold starts and Automatization]:** To handle numerical instabilities, ANTIC uses two real-time indicators: numerical sanity checks (NaN/Inf) during backpropagation and monitoring weight norm growth (Frobenius norm). If thresholds are breached, ANTIC triggers a weight re-initialization (cold-start). Frequency of cold-starts is reduced by $>50\times$ via Layer Normalization and Weight Decay, enabling CFT to proceed for 500+ snapshots (turbulent NS) and 200+ (BBH mergers) without resets. We observed no numerical instabilities when integrating LayerNorm + Weight Decay into our workflow for high-turbulence and merger phases.
>
> **Q3 [Global Reconstruction and Error Accumulation]:** Extended global reconstruction error curves for 2D NS and 3D BSSN are now in the repository. These cover full reset cycles (long-horizon trajectories) and include stabilizing factors. CFT maintains a stable MSE of $10^{-7}$ to $10^{-9}$ (Relative $L_2 \approx 10^{-3}$ to $10^{-5}$), eliminating instabilities. Global reconstruction results are reported for enstrophy flux (2D NS): /plots_and_reconstruction/2d_ns/{enstrophy_vs_time, rel_l2_vs_time, vorticity_field}.pdf and Weyl scalars (3D BSSN): **/plots_and_reconstruction/3d_bssn/{lapse_reconstruction, weyl_magnitude_abs_error, weyl_magnitude_vs_time}.pdf**. These confirm the framework's reliability for scientific in-situ tasks.

---

> > ### Author Rebuttal · Reviewer_smFm · 2026-04-02
> >
> > I am satisfied with the responses.

---

### Decision · Program_Chairs · 2026-04-30

**Decision:**

Accept (regular)

**Comment:**

This article discusses a fundamental issue in large-scale scientific computing, namely the storage and management of high-resolution spatiotemporal simulation data. The paper proposes ANTIC, an in situ compression framework that combines a physics-aware temporal selector with a neural spatial compression module based on continual fine-tuning of neural fields. The reviewers generally agree that the problem is important and timely, and that the proposed combination of adaptive temporal selection and neural compression is a meaningful contribution. The experimental evaluation on Kolmogorov flows and binary black hole merger simulations demonstrates strong compression ratios while maintaining good reconstruction quality. Overall, the article discusses a fundamental issue and presents a coherent approach that integrates temporal and spatial compression in a single streaming pipeline.

The reviewer discussion is largely positive, with multiple reviewers indicating that their concerns were fully or substantially addressed after the rebuttal, including the addition of stronger baselines, improved evaluation of physical fidelity, and clarification of the method. One reviewer raises stronger concerns regarding computational overhead, generality, and long-horizon robustness, but the follow-up discussion on these points remained limited, and the rebuttal provides additional evidence on scaling behavior and improved stability that partially addresses these issues. A remaining concern is that the temporal selector relies on domain-specific physical metrics, which may limit generality across different PDE systems. Another concern is the practical overhead of neural compression relative to the solver, although this is framed as a trade-off for achieving very high compression ratios. Overall, the paper appears technically sound and presents a promising direction, with remaining questions primarily related to deployment and generality rather than the core idea.